# Kermut: Composite kernel regression for protein variant effects

**Peter Mørch Groth**[*][†]
University of Copenhagen
Novonesis

**Mads Herbert Kerrn**[*][†]
University of Copenhagen

**Lars Olsen**
Novonesis

**Jesper Salomon**
Novonesis

**Wouter Boomsma**[†]
University of Copenhagen

## Abstract

Reliable prediction of protein variant effects is crucial for both protein optimization and for advancing biological understanding. For practical use in protein engineering, it is important that we can also provide reliable uncertainty estimates for our predictions, and while prediction accuracy has seen much progress in recent years, uncertainty metrics are rarely reported. We here provide a Gaussian process regression model, Kermut, with a novel composite kernel for modeling mutation similarity, which obtains state-of-the-art performance for supervised protein variant effect prediction while also offering estimates of uncertainty through its posterior. An analysis of the quality of the uncertainty estimates demonstrates that our model provides meaningful levels of overall calibration, but that instance-specific uncertainty calibration remains more challenging.

## 1 Introduction

Accurately predicting protein variant effects is crucial for both advancing biological understanding and for engineering and optimizing proteins towards specific traits. Recently, much progress has been made in the field as a result of advances in machine learning-driven modeling [1–3], data availability [4, 5], and relevant benchmarks [6, 7].

While prediction accuracy has received considerable attention, the ability to quantify the uncertainties of predictions has been less intensely explored. This is of immediate practical consequence. One of the main purposes of protein variant effect prediction is as an aid for protein engineering and design, to propose promising candidates for subsequent experimental characterization. For this purpose, it is essential that we can quantify, on an instance-to-instance basis, how trustworthy our predictions are. Specifically, in a Bayesian optimization setting, most choices of acquisition function actively rely on predicted uncertainties to guide the optimization, and well-calibrated uncertainties have been shown to correlate with optimization performance [8].

Our goal with this paper is to start a discussion on the quality of the estimated uncertainties of supervised protein variant effect prediction. Gaussian Processes (GP) are a standard choice for uncertainty quantification due to the closed form expression of the posterior. We therefore first ask the question whether state-of-the-art performance can be obtained within the GP framework. We propose a composite kernel that comfortably achieves this goal, and subsequently investigate the quality of the uncertainty estimates from such a model. Our results show that while standard approaches like reliability diagrams give the impression of good levels of calibration, the quantification of per-instance uncertainties is more challenging. We make our model available as a baseline and encourage the

---

[*]equal contribution
[†]corresponding authors: `petergroth@di.ku.dk, make@di.ku.dk, wb@di.ku.dk`

38th Conference on Neural Information Processing Systems (NeurIPS 2024).

community to place greater emphasis on uncertainty quantification in this important domain. Our contributions can be summarized as follow:

- We introduce **Kermut**, a Gaussian process with a novel composite **ker**nel for modeling **mut**ation similarity, leveraging signals from pretrained sequence and structure models;

- We evaluate this model on the comprehensive ProteinGym substitution benchmark and show that it is able to reach state-of-the-art performance in supervised protein variant effect prediction, outperforming recently proposed deep learning methods in this domain;

- We provide a thorough calibration analysis and show that while Kermut provides well-calibrated uncertainties overall, the calibratedness of instance-specific uncertainties remains challenging;

- We demonstrate that our model can be trained and evaluated orders of magnitude faster and with better out-of-the-box calibration than competing methods.

## 2 Related work

### 2.1 Protein property prediction

Predicting protein function and properties using machine-learning based approaches continues to be an innovative and important area of research.

Recently, *unsupervised* approaches have gained significant momentum where models trained in a self-supervised fashion have shown impressive results for zero-shot estimates of protein *fitness* and *variant effects* relative to a reference protein [3, 9–11].

*Supervised* learning is a crucial method of utilizing experimental data to predict protein fitness. This is particularly valuable when the trait of interest correlates poorly with the evolutionary signals that unsupervised models capture during training or if multiple traits are considered. Supervised protein fitness prediction using machine learning has been explored in detail in [12], where a comprehensive overview can be found. A common strategy is to employ transfer learning via embeddings extracted from self-supervised models [13, 14], an approach which increasingly relies on large pretrained language models such as ProtTrans [15], ESM-2 [16], and SaProt [17]. In [18], the authors propose to augment a one-hot encoding of the aligned amino acid sequence by concatenating it with a zero-shot score for improved predictions. This was further expanded upon with ProteinNPT [19], where sequences embedded with the MSA Transformer [20] and zero-shot scores were fed to a transformer architecture for state-of-the-art supervised variant effect prediction with generative capabilities.

Considerable progress has been made in defining meaningful and comprehensive benchmarks to reliably measure and compare model performance in both unsupervised and supervised protein fitness prediction settings. The FLIP benchmark [6] introduced three supervised predictions tasks ranging from local to global fitness prediction, where each task in turn was divided into clearly defined splits. The supervised benchmarks often view fitness prediction through a particular lens. Where FLIP targeted problems of interest to protein engineering; TAPE [21] evaluated transfer learning abilities; PEER [22] focused on sequence understanding; ATOM3D [23] considered a structure-based approach; FLOP [24] targeted wild type proteins; and ProteinGym focused exclusively on variant effect prediction [11]. The ProteinGym benchmark was recently expanded to encompass more than 200 standardized datasets in both zero-shot and supervised settings, including substitutions, insertions, deletions, and curated clinical datasets [11, 7].

### 2.2 Kernel methods for protein sequences

Kernel methods have seen much use for protein modeling and protein property prediction. Sequence-based string kernels operating directly on the protein amino acid sequences are one such example, where, e.g., matching $k$-mers at different $k$s quantify covariance. This has been used with support vector machines to predict protein homology [25, 26]. Another example is sub-sequence string kernels, which in [27] is used in a Gaussian process for a Bayesian optimization procedure. In [28], string kernels were combined with predicted physicochemical properties to improve accuracy in the prediction of MHC-binding peptides and in protein fold classification. In [29], a kernel leveraging the tertiary structure for a protein family represented as a residue-residue contact map was used to

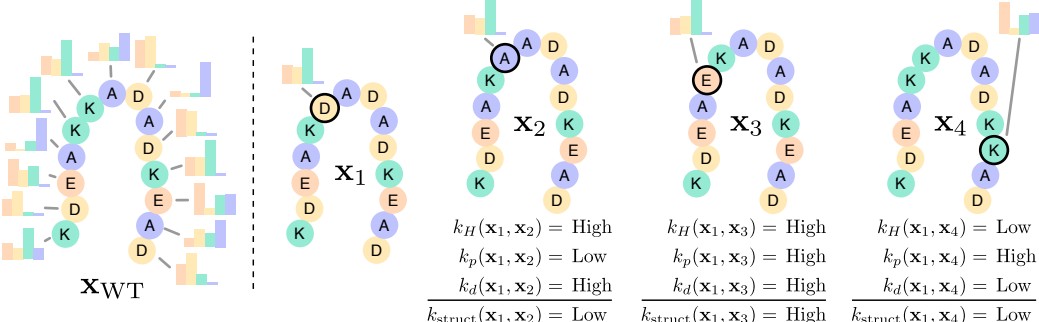

Figure 1: Overview of Kermut's structure kernel. Using an inverse folding model, structure-conditioned amino acid distributions are computed for all sites in the reference protein. The structure kernel yields high covariances between two variants if the local environments are similar, if the mutation probabilities are similar, and if the mutates sites are physically close. Constructed examples of expected covariances between variant $\mathbf{x}_1$ and $\mathbf{x}_{2,3,4}$ are shown.

predict various protein properties such as enzymatic activity and binding affinity. In [30], Gaussian process regression (GPR) was used to successfully identify promising enzyme sequences which were subsequently synthesized showing increased activity. In [31], the authors provide a comprehensive study of kernels on biological sequences which includes a thorough review of the literature as well as both theoretical, simulated, and in-silico results.

Most similar to our work is mGPfusion [32], in which a weighted decomposition kernel was defined which operated on the local tertiary protein structure in conjunction with a number of substitution matrices. Simulated stability data for all possible single mutations were obtained via Rosetta [33], which was then fused with experimental data for accurate $\Delta\Delta$G predictions of single- and multi-mutant variants via GPR, thus incorporating both sequence, structure, and a biophysical simulator. In contrast to our approach, the mGPfusion-model does not leverage pretrained models, but instead relies on substitution matrices for its evolutionary signal. A more recent example of kernel-based methods yielding highly competitive results is xGPR [34], in which Gaussian processes with custom kernels show high performance when trained on protein language model embeddings, similarly to the sequence kernel in our work (see Section 3.3). Where xGPR introduces a set of novel random feature-approximated kernels with linear-scaling, Kermut instead uses the squared exponential kernel for sequence modeling while additionally modeling local structural environments. The models in xGPR were shown to provide both high accuracy and well-calibrated uncertainty estimation on the FLIP and TAPE benchmarks.

### 2.3 Uncertainty quantification and calibration

Uncertainty quantification (UQ) for protein property prediction continues to be a promising area of research with immediate practical consequences. In [35], residual networks were used to model both epistemic and aleatoric uncertainty for peptide selection. In [36], GPR on MLP-residuals from biLSTM embeddings was used to successfully guide in-silico experimental design of kinase binders and fluorescent proteins. The authors of [37] augmented a Bayesian neural network by placing biophysical priors over the mean function by directly using Rosetta energy scores, whereby the model would revert to the biophysical prior when the epistemic uncertainty was large. This was used to predict fluorescence, binding, and solubility for drug-like molecules. In [38], state-of-the-art performance on protein-protein interactions was achieved by using a spectral-normalized neural Gaussian process [39] with an uncertainty-aware transformer-based architecture working on ESM-2 embeddings.

In [40], a framework for evaluating the epistemic uncertainty of deep learning models using confidence interval-based metrics was introduced, while [41] conducted a thorough analysis of uncertainty quantification methods for molecular property prediction. Here, the importance of supplementing confidence-based calibration with error-based calibration as introduced in [42] was highlighted, whereby the predicted uncertainties are connected directly to the expected error for a more nuanced calibration analysis. We evaluate our model using confidence-based calibration as well as error-based

calibration following the guidelines in [41]. In [43], the authors conducted a systematic comparison of UQ methods on molecular property regression tasks, while [44] investigated calibratedness of regression models for material property prediction. In [45], the above approaches were expanded to protein property prediction tasks where the FLIP [6] benchmark was examined, while [46] benchmarked a number of UQ methods for molecular representation models. In [47], the authors developed an active learning approach for partial charge prediction of metal-organic frameworks via Monte Carlo dropout [48] while achieving decent calibration. In [49], a systematic analysis of protein regression models was conducted where well-calibrated uncertainties were observed for a range of input representations.

## 2.4 Local structural environments

Much work has been done to solve the inverse-folding problem, where the most probable amino acid sequence to fold into a given protein backbone structure is predicted [50–56]. Inverse-folding models are trained on large datasets of protein structures and model the physicochemical and evolutionary constraints of sites in a protein conditioned on their structural contexts. These will form the basis of the structural featurization in our work. Local structural environments have previously been used for protein modeling. In [57], a 3D CNN was used to predict amino acid preferences giving rise to novel substitution matrices. In [58] and [59], surface-level fingerprinting given structural environments was used to model protein-protein interaction sites and for de novo design of protein interactions. In [60], chemical microenvironments were used to identify potentially beneficial mutations, while [61] used a similar approach, where they investigated the volume of the local environments and observed that the first contact shell delivered the primary signal thus emphasizing the importance of locality. In [62], a composition Hellinger distance metric based on the chemical composition of local residue environments was developed and used for a range of structure-related experiments. Recently, local structural environments were used to model mutational preferences for protein engineering tasks [63], however not in a Gaussian process framework as we propose here.

# 3 Methods

## 3.1 Preliminaries

We want to predict the outcome of an assay measured on a protein, represented by its amino acid sequence $\mathbf{x}$ of length $L$. We will assume that we have a dataset of $N$ such sequences available, and that these are of equal length and structure, such that we can meaningfully refer to the effect at specific positions (sites) in the protein. In protein engineering, we typically consider modifications relative to an initial wild type sequence, $\mathbf{x}_{\text{WT}}$. We will assume that the 3D structure for the initial sequence, $s$, is available (either experimentally determined or provided by a structure predictor like AlphaFold [64]). Lastly, for variant $\mathbf{x}$ with mutations at sites $M \subseteq \{1, ..., L\}$, let $\mathbf{x}^m$ denote the variant which has the same mutation as $\mathbf{x}$ at site $m$ for $m \in M$ and otherwise is equal to $\mathbf{x}_{\text{WT}}$.

## 3.2 Gaussian processes

To predict protein variant effects, we rely on Gaussian process regression, which we shall now briefly introduce. For a comprehensive overview, see [65], which this section is based on.

Let $\mathscr{X}$ and $\mathscr{Y}$ be two random variables on the measurable spaces $\mathcal{X}$ and $\mathbb{R}$, respectively, and let $X = \mathbf{x}_1, ..., \mathbf{x}_N$ and $\mathbf{y} = y_1, ..., y_N$ be realizations of these random variables. We assume that $y_i = g(\mathbf{x}_i) + \epsilon$, where $g$ represents some unknown function and $\epsilon \sim \mathcal{N}(0, \sigma_\epsilon^2)$ accounts for random noise. Our objective is to model the distributions capturing our belief about $g$.

Gaussian processes are stochastic processes providing a powerful framework for modeling distributions over functions. The Gaussian process framework allows us to not only make predictions but also to quantify the uncertainty associated with each prediction. A Gaussian process is entirely specified by its *mean* and *covariance* functions, $m(\mathbf{x})$ and $k(\mathbf{x}, \mathbf{x}')$. We assume that the covariance matrix, $K$, of the outputs $\{y_1, ..., y_N\}$ can be parameterized by a function of their inputs $\{\mathbf{x}_1, ..., \mathbf{x}_N\}$. The parameterization is defined by the kernel, $k : \mathcal{X} \times \mathcal{X} \to \mathbb{R}$ yielding $K$ such that $K_{ij} = k(\mathbf{x}_i, \mathbf{x}_j)$. For $k$ to be a valid kernel, $K$ needs to be symmetric and positive semidefinite.

Let $f$ represent our approximation of $g$, $f(\mathbf{x}) \approx g(\mathbf{x})$. Given a training set $\mathcal{D} = (X, \mathbf{y})$ and a number of test points $X_*$, the function $\mathbf{f}_*$ predicts the values of $y_*$ at $X_*$. Using rules of normal distributions, we derive the posterior distribution $p(\mathbf{f}_* | X_*, \mathcal{D})$, providing both a prediction of $y_*$ at $X_*$ and a confidence measure, often expressed as $\pm 2\sigma$, where $\sigma$ is the posterior standard deviation at a test point $\mathbf{x}_*$.

The kernel function often contains hyperparameters, $\eta$. These can be optimized by maximizing the marginal likelihood, $p(\mathbf{y} | X, \eta)$, which is known as type II maximum likelihood.

### 3.3 Kermut

It has been shown that local structural dependencies are useful for determining mutation preferences [57, 60, 61, 63]. We therefore hypothesize that constructing a composite kernel with components incorporating information about the local structural environments of residues will be able to model protein variant effects. To this end, we define a structure kernel, $k_{\text{struct}}$, which models mutation similarity given the local environments of mutated sites. A schematic of how the structure kernel models covariances can be seen in Figure 1. In the following we shall define $k_{\text{struct}}^1$, a structure kernel that operating on single-mutant variants. Subsequently, we shall extend it to multi-mutant variants resulting in the structure kernel, $k_{\text{struct}}$.

We hypothesize that for a given site in a protein, the distribution over amino acids given by a structure-conditioned inverse folding model will reflect the effect of a mutation at that site. We consider such an amino acid distribution a representation of the *local environment* for that site as it reflects the various physicochemical and evolutionary constraints that the site is subject to. We thus presume that two sites with similar local environments will behave similarly if mutated. For instance, mutations at buried sites in the hydrophobic core of the protein will generally correlate more with each other than with surface-level mutations.

For single mutant variants we quantify site similarity using the Hellinger kernel $k_H(\mathbf{x}, \mathbf{x}') = \exp(-\gamma_1 d_H(f_{\text{IF}}(\mathbf{x}), f_{\text{IF}}(\mathbf{x}')))$, with $\gamma_1 > 0$ [66], where $d_H$ is the Hellinger distance (see Appendix B.1). The function $f_{\text{IF}} : \mathcal{X}^1 \to [0, 1]^{20}$ takes a single-mutant sequence, $\mathbf{x}$, as input and returns a probability distribution over the 20 naturally occurring amino acids at the mutated site in $\mathbf{x}$ given by the inverse folding model. The Hellinger kernel will assign maximum covariance when two sites are identical. This however means that $k_H$ is incapable of distinguishing between different mutations at the same site since $d_H(\mathbf{x}, \mathbf{x}') = 1$, when $\mathbf{x}$ and $\mathbf{x}'$ are mutated at the same site.

To increase flexibility and to allow intra-site comparisons, we introduce a kernel operating on the specific mutation likelihoods. We hypothesize that two variants with mutations on sites that are close in terms of the Hellinger distance will correlate further if the log-probabilities of the specific amino acids on the mutated sites are similar (i.e., the probability of the amino acid that we mutate *to* is similar at the two sites). We incorporate this by defining $k_p(\mathbf{x}, \mathbf{x}') = k_{\text{exp}}(f_{\text{IF}_1}(\mathbf{x}), f_{\text{IF}_1}(\mathbf{x}')) = \exp(-\gamma_2 || f_{\text{IF}_1}(\mathbf{x}) - f_{\text{IF}_1}(\mathbf{x}') ||)$, where $f_{\text{IF}_1} : \mathcal{X}^1 \to [0, 1]$ takes a single-mutant sequence, $\mathbf{x}$, as input and returns the log-probability (given by an inverse folding model) of the observed mutation, and where $k_{\text{exp}}$ is the exponential kernel.

Finally, we hypothesize that the effect of two mutations correlate further if the sites are close in physical space. Hence, we multiply the kernel with an exponential kernel on the Euclidean distance between sites: $k_d(\mathbf{x}, \mathbf{x}') = \exp(-\gamma_3 d_e(s_i, s_j))$. Thereby, the closer two sites are physically, the more similar – and thus comparable – their local environments will be.

Taking the product of these kernel components, we get the following kernel for single-mutant variants, which assigns high covariance when two single mutant variants have mutations that have similar environments, are physically close, and have similar mutation likelihoods:

$$k_{\text{struct}}^1(\mathbf{x}, \mathbf{x}') = \lambda k_H(\mathbf{x}, \mathbf{x}') k_p(\mathbf{x}, \mathbf{x}') k_d(\mathbf{x}, \mathbf{x}'), \tag{1}$$

where the kernel has been scaled by a non-negative scalar, $\lambda > 0$.

In [63], the authors showed that a simple linear model operating on one-hot encoded mutations is sufficient to accurately predict mutation effects given sufficient data. Thus, we generalize the kernel to multiple mutations by summing over all pairs of sites differing at $\mathbf{x}$ and $\mathbf{x}'$:

$$k_{\text{struct}}(\mathbf{x}, \mathbf{x}') = \sum_{i \in M} \sum_{j \in M'} k_{\text{struct}}^1(\mathbf{x}^i, \mathbf{x}'^j) \tag{2}$$

Table 1: Performance on the ProteinGym benchmark. Best results are bold. Kermut reaches superior performance across splits with significant gains in the challenging modulo and contiguous settings. OHE and NPT model types correspond to one-hot encodings and non-parametric transformers.

| Model type | Model name | Spearman ($\uparrow$) | | | | MSE ($\downarrow$) | | | |
|---|---|---|---|---|---|---|---|---|---|
| | | Contig. | Mod. | Rand. | Avg. | Contig. | Mod. | Rand. | Avg. |
| OHE | None | 0.064 | 0.027 | 0.579 | 0.224 | 1.158 | 1.125 | 0.898 | 1.061 |
| | ESM-1v | 0.367 | 0.368 | 0.514 | 0.417 | 0.977 | 0.949 | 0.764 | 0.897 |
| | DeepSequence | 0.400 | 0.400 | 0.521 | 0.440 | 0.967 | 0.940 | 0.767 | 0.891 |
| | MSAT | 0.410 | 0.412 | 0.536 | 0.453 | 0.963 | 0.934 | 0.749 | 0.882 |
| | TranceptEVE | 0.441 | 0.440 | 0.550 | 0.477 | 0.953 | 0.914 | 0.743 | 0.870 |
| Embed. | ESM-1v | 0.481 | 0.506 | 0.639 | 0.542 | 0.937 | 0.861 | 0.563 | 0.787 |
| | MSAT | 0.525 | 0.538 | 0.642 | 0.568 | 0.836 | 0.795 | 0.573 | 0.735 |
| | Tranception | 0.490 | 0.526 | 0.696 | 0.571 | 0.972 | 0.833 | 0.503 | 0.769 |
| NPT | ProteinNPT | 0.547 | 0.564 | 0.730 | 0.613 | 0.820 | 0.771 | 0.459 | 0.683 |
| GP | Kermut | **0.610** | **0.633** | **0.744** | **0.662** | **0.699** | **0.652** | **0.414** | **0.589** |

The structure kernel models mutations linearly and cannot capture epistatic effects. We propose to add epistatic signals through a sequence kernel. Drawing on the rich literature for modeling protein sequences with Gaussian processes on embeddings [34, 45, 49, 67], we use a squared exponential kernel which operates on sequence embeddings from a pretrained model. We use the 650M parameter ESM-2 protein language model [16], and perform mean-pooling across the length dimension, yielding $\mathbf{z} = f_1(\mathbf{x})$, where $f_1$ produces mean-pooled embeddings, $\mathbf{z} \in \mathbb{R}^{1280}$, of sequence $\mathbf{x}$. We thus model the covariance between these representations as

$$k_{\text{seq}}(\mathbf{x}, \mathbf{x}') = k_{\text{SE}}(f_1(\mathbf{x}), f_1(\mathbf{x}')) = k_{\text{SE}}(\mathbf{z}, \mathbf{z}') = \exp\left(-\frac{||\mathbf{z} - \mathbf{z}'||_2^2}{2\sigma^2}\right). \tag{3}$$

We choose to add and weigh the structure and sequence kernels resulting in our final kernel formulation, whereby the model can leverage either structure or sequence similarities, depending on the presence and strength of each signal as determined through hyperparameter optimization:

$$k(\mathbf{x}, \mathbf{x}') = \pi k_{\text{struct}}(\mathbf{x}, \mathbf{x}') + (1 - \pi)k_{\text{seq}}(\mathbf{x}, \mathbf{x}'). \tag{4}$$

Additional details on both sequence and structure kernels, including a proof of the validity of the structure kernel, implementation details, computational complexity details, and an automatic model selection procedure can be found in Appendices B and C.

### 3.3.1 Zero-shot mean function

Kermut can be used with a constant mean function, $m(\mathbf{x}) = \alpha$, where $\alpha$ is a hyperparameter optimized through the marginal likelihood. However, we posit that additional performance can be gained by using an altered mean function which operates on zero-shot fitness estimates, which are often available at relatively low cost: $m(\mathbf{x}) = \alpha f_0(\mathbf{x}) + \beta$, where $f_0$ is a zero-shot method evaluated on input sequence $\mathbf{x}$. This is similar to the approach employed in [37], where Rosetta scores are used as a biophysical prior. We use ESM-2 [16], which yields the log-likelihood ratio between the variant and wild type residue as in [10]. For details, see Appendix B.

### 3.4 Architecture considerations

While Kermut is based on relatively simple principles, its components are non-trivial in that they exploit learned features from pretrained models: ESM-2 provides protein sequence level embeddings and zero-shot scores, while ProteinMPNN provides a featurization of the local structural environments. We stress that these pretrained components can be readily replaced by other pretrained models. Models that generate (1) protein sequence embeddings, (2) structure-conditioned amino acid distributions, and (3) zero-shot scores are plentiful and such models will progress further in future years. In our work, we have not sought to find the optimal combination of these. Our work should instead be seen

Table 2: Ablation results. Key components of the kernel are removed and the model is trained and evaluated on 174/217 assays from the ProteinGym benchmark. The ablation column shows the alteration to the GP formulation. The metrics are subtracted from Kermut to show the change in performance. Negative $\Delta$Spearman values indicate a drop in performance.

| Description | Ablation | $\Delta$Spearman | | | |
| --- | --- | --- | --- | --- | --- |
| | | Contig. | Mod. | Rand. | Avg. |
| No structure kernel | $k_{\text{struct}} = 0$ | $-0.082$ | $-0.078$ | $-0.033$ | $-0.065$ |
| No sequence kernel | $k_{\text{seq}} = 0$ | $-0.040$ | $-0.046$ | $-0.048$ | $-0.045$ |
| No inter-residue dist. | $k_d = 1$ | $-0.049$ | $-0.051$ | $-0.004$ | $-0.035$ |
| No mut. prob./site comp. | $k_p = k_H = 1$ | $-0.035$ | $-0.037$ | $-0.006$ | $-0.026$ |
| Const. mean | $m(\mathbf{x}) = \alpha$ | $-0.036$ | $-0.032$ | $-0.008$ | $-0.026$ |
| No mut. prob. | $k_p = 1$ | $-0.022$ | $-0.019$ | $-0.005$ | $-0.016$ |
| No site comp. | $k_H = 1$ | $-0.006$ | $-0.009$ | $0.000$ | $-0.006$ |

as a framework demonstrating how such components can be meaningfully combined in a GP setting to obtain state-of-the-art results.

## 4 Results

We evaluate Kermut on the 217 substitution DMS assays from the ProteinGym benchmark [7]. The overall benchmark results are an aggregate of three different cross-validation schemes: In the "random" scheme, variants are assigned to one of five folds randomly. In the "modulo" scheme, every fifth position along the protein backbone are assigned to the same fold, and in the "contiguous" scheme, the protein is split into five equal-sized segments along its length, each constituting a fold. For all three schemes, models are trained on four combined partitions and tested on the fifth for a total of five runs per assay, per scheme. The results are processed using the functionality provided in the ProteinGym repository. The average and per-scheme aggregated results can be seen in Table 1.

Our model reaches both higher Spearman correlations and lower mean squared errors than competing methods and thereby achieves state-of-the-art performance in all three schemes, with the largest gains in the challenging modulo and contiguous settings. In Table E.2, the performance per functional category is shown, where we observe a significant performance increase in binding- and stability-related prediction tasks, likely explained by inclusion of the structure kernel. In addition to its high accuracy, Kermut is significantly faster compared to deep learning methods. We provide wall-clock times for running a 5-fold CV loop for a single split-scheme for four select datasets for both Kermut and ProteinNPT in Table C.1. Generating results for all three split schemes for the four datasets thus takes Kermut approximately 10 minutes while ProteinNPT takes upwards of 400 hours.

The non-parametric bootstrap standard error for each model relative to Kermut can be seen in Tables E.1 and E.3. In Appendix M, we provide additional results using alternate zero-shot functions. The average and per-split performance for individual assays can be seen in Figures O.1 to O.4 while additional details on computation time can be seen in Appendix C. Lastly, visualizations of the distributions of optimized hyperparameters can be seen in Appendix N.

### 4.1 Ablation study

To examine the impact of Kermut's components, we conduct an ablation study, where we ablate its two main kernels from Equation (4) – the structure and sequence kernels – as well as the structure kernel's subcomponents. We similarly investigate the importance of the zero-shot mean function. The ablation study is carried out on all split schemes on a subset of 174 datasets. The difference between the ablation results and the Kermut results can be seen in Table 2, where larger values indicate large component importance. For the absolute values, see Appendix F, where we additionally include alternative kernel formulations for both structure and sequence kernels as well as for kernel composition.

As indicated by the largest drop in performance, the single most important component is the structure kernel. While removing the sequence kernel leads to comparable decreases for all three schemes,

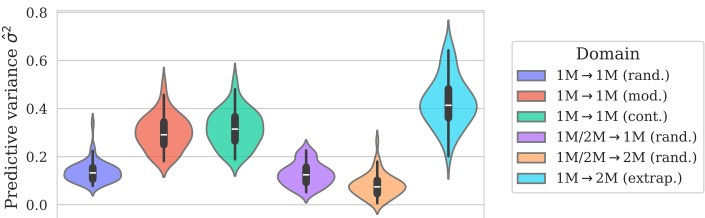

Figure 2: Distribution of predictive variances for datasets with double mutants, grouped by domain. The three first elements correspond to the three split-schemes from ProteinGym. The third and fourth correspond to training on both single and double mutants, and testing on each, respectively. For the last column, we train on single and test on double mutants, corresponding to an extrapolation setting.

removing the structure kernel primarily leads to drops in the challenging contiguous and modulo schemes. This shows that the structure kernel is crucial for characterizing unseen sites in the wild type protein. While removing the site comparison and mutation probability kernels leads to small and medium drops in performance, we observe that removing both leads to an even larger performance drop, indicating a synergy between the two. In Table E.2, the ablation results per functional category are shown, where observe that the inclusion of the structure kernel is crucial for the increased performance in structure-related prediction tasks such as binding and stability.

## 4.2 Uncertainty quantification per mutation domain

By inspecting the posterior predictive variance, we can analyze model uncertainty. To this end, we define several *mutation domains* of interest [49]. We designate the three split schemes from the ProteinGym benchmark as three such domains. These are examples of interpolation, where we both train and test on single mutants (1M→1M). While the main benchmark only considers single mutations, some assays include additional variants with multiple mutations. We consider a number of these and define two additional interpolation domains where we train on *both* single and double mutations (1M/2M) and test on singles and doubles, respectively. As a challenging sixth domain, we train on single mutations only and test on doubles (1M→2M), constituting an extrapolation domain. For details on the multi-mutant splits, see Appendix G.

Figure 2 shows the distributions of mean predictive variances in the six domains. In the three single mutant domains, we observe that the uncertainties increase from scheme to scheme, reflecting the difficulties of the tasks and analogously the expected performance scores (Table 1). When training on both single and double mutants (1M/2M), we observe a lower uncertainty on double mutants than single mutants. For many of the multi-mutant datasets, the mutants are not uniformly sampled but often include a fixed single mutation. A possible explanation is thus that it might be more challenging to decouple the signal from a double mutation into its constituent single mutation signals. In the extrapolation setting, we observe large predictive uncertainties, as expected. One explanation of the discrepancy between the variance distributions in the multi-mutant domains might lie in the difference in target distributions between training and test sets. Figure I.1 in the appendix shows the overall target distribution of assays for the 51 considered multi-mutant datasets. The single and double mutants generally belong to different modalities, where the double mutants often lead to a loss of fitness. This shows the difficulty of predicting on domains not encountered during training. For reference, we include the results for the multi-mutant domains in Table G.1 in the appendix.

## 4.3 Uncertainty calibration analysis

To clarify the relationship between model uncertainty and expected performance, we proceed with a calibration analysis. First, we perform a confidence interval-based calibration analysis [41], resulting in calibration curves which in the classification setting are known as reliability diagrams [68]. The results for each dataset are obtained via five-fold cross validation, corresponding to five separately trained models for each split scheme. We select four diverse datasets as examples (Table 3), reflecting both high and low predictive performance. The mean calibration curves can be seen in Figure 3a. For method details and results across all datasets, see Appendix J. The mean *expected calibration error* (ECE) is shown in the bottom of each plot, where a value of zero indicates perfect calibration. Overall,

Table 3: Details and results for four diverse ProteinGym datasets used for calibration analysis. The results show the Spearman correlation for each CV-scheme and the average correlation.

| Uniprot ID | Spearman ($\uparrow$) | | | | Details | | | |
|---|---|---|---|---|---|---|---|---|
| | Contig. | Mod. | Rand. | Avg. | $N$ | $L$ | Assay | Source |
| BLAT_ECOLX | 0.804 | 0.826 | 0.909 | 0.846 | 4996 | 286 | Organismal fitness | [69] |
| PA_I34A1 | 0.226 | 0.457 | 0.539 | 0.407 | 1820 | 716 | Organismal fitness | [70] |
| TCRG1_MOUSE | 0.849 | 0.849 | 0.928 | 0.875 | 621 | 37 | Stability | [4] |
| OPSD_HUMAN | 0.739 | 0.734 | 0.727 | 0.734 | 165 | 348 | Expression | [71] |

the uncertainties appear to be well-calibrated both qualitatively from the curves and quantitatively from the ECEs. Even the smallest dataset (fourth row, $N = 165$) achieves decent calibration, albeit with larger variances between folds.

While the confidence interval-based calibration curves show that we can trust the uncertainty estimates overall, they do not indicate whether we can trust individual predictions. We therefore supplement the above analysis with an error-based calibration analysis [42], where a well-calibrated model will have low uncertainty when the error is low. The calibration curves can be seen in Figure 3b. We compute the per CV-fold *expected normalized calibration error* (ENCE) and the *coefficient of variation* ($c_v$), which quantifies the variance of the predicted uncertainties. Ideally, the ENCE should be zero while the coefficient of variation should be relatively large (indicating spread-out uncertainties).

While the confidence interval-based analysis suggested that the uncertainty estimates are well-calibrated with some under-confidence, the same is not as visibly clear for the error-based calibration plots, suggesting that the expected correlation between model uncertainty and prediction error is not a given. We do however see an overall trend of increasing error with increasing uncertainty in three of the four datasets, where the curves lie close to the diagonal (as indicated by the dashed line). The second row shows poorer calibration – particularly in the modulo and contiguous schemes. The curves however remain difficult to interpret, in part due to the errorbars on both axes. To alleviate this, we compute similar metrics for all 217 datasets across the three splits, which indicate that, though varying, most calibration curves are well-behaved (see Appendix J.3).

We supplement the above calibration curves with Figures K.1 to K.4 in the Appendix, which show the true values plotted against the predictions. These highlight the importance of well-calibrated uncertainties and underline their role in interpreting model predictions and their trustworthiness.

### 4.3.1 Comparison with ProteinNPT

Monte Carlo (MC) dropout [48] is a popular uncertainty quantification technique for deep learning models. Calibration curves for ProteinNPT with MC dropout for the same four datasets across the three split schemes can be seen in Appendix L.2 while figures showing the true values plotted against the predictions with uncertainties are shown in Appendix L.3. These indicate that employing MC dropout on a deep learning model like ProteinNPT seems to provide lower levels of calibration, providing overconfident uncertainties across assays and splits. Due to the generally low uncertainties and the resulting difference in scales, the calibration curves are often far from the diagonal. The trends in the calibration curves however show that the model errors often correlate with the uncertainties, suggesting that the model can be recalibrated to achieve decent calibration.

Other techniques for uncertain quantification in deep learning models certainly exist, and we by no means rule out that other techniques can outperform our method (see Discussion below). We note, however, that many uncertainty quantification methods will be associated with considerable computational overhead compared to the built-in capabilities of a Gaussian process.

## 5   Discussion

We have shown that a carefully constructed Gaussian process is able to reach state-of-the-art performance for supervised protein variant effect prediction while providing reasonably well-calibrated

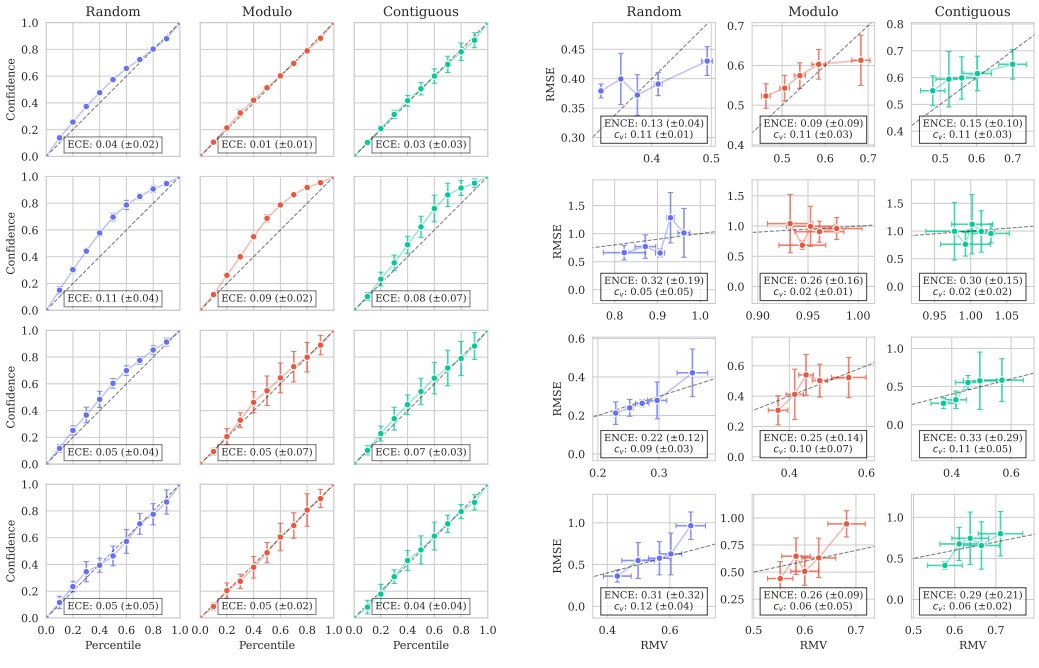

(a) Confidence interval-based calibration curves.

(b) Error-based calibration curves.

Figure 3: Calibration curves for Kermut using different methods. Mean ECE/ENCE values ($\pm 2\sigma$) are shown. Dashed line ($x = y$) corresponds to ideal calibration. The row order corresponds to the ordering in Table 3. (a) exhibits good calibration as indicated by curves close to the diagonal and ECE values close to zero, albeit with under-confident uncertainties in the second row. In (b), Kermut is also relatively well-calibrated, as indicated by the increasing curves, albeit with large variances along both axes. The low coefficients of variation ($c_v$) indicate similar predictive variances in each setting. Overall, Kermut achieves good calibration in most cases as a result of the designed kernel.

uncertainty estimates. For a majority of datasets, this is achieved orders of magnitude faster than competing methods.

While the predictive performance on the substitution benchmark is an improvement over previous methods, our proposed model has its limitations. Due to the site-comparison mechanism, our model is unable to handle insertions and deletions as it only operates on a fixed structure. Additionally, as the number of mutations increases, the assumption of a fixed structure might worsen, depending on the introduced mutations, which can affect reliability as the local environments might change. An additional limitation is the GP's $\mathcal{O}(N^3)$ scaling with dataset size. While not a major obstacle in the single mutant setting, dataset sizes can quickly grow when handling multiple mutants. The last decades have however produced a substantial literature on algorithms for scaling GPs to larger datasets [72–74], which could alleviate the issue, and we therefore believe this to be a technical rather than fundamental limitation. An additional limitation might present itself it in the multi-mutant setting, where the lack of explicit modeling of epistasis can potentially hinder extrapolation to higher-order mutants, prompting further investigation.

Well-calibrated uncertainties are crucial for protein engineering; both when relying on a Bayesian optimization routine to guide experimental design using uncertainty-dependent acquisition functions and similarly to weigh the risk versus reward for experimentally synthesizing suggested variants. We therefore encourage the community to place a greater emphasis on uncertainty quantification and calibration for protein prediction models as this will have measurable impacts in real-life applications like protein engineering – perhaps more so than increased prediction accuracy. We hope that Kermut can serve as a fruitful step in this direction.

## Acknowledgments and Disclosure of Funding

This work was funded in part by Innovation Fund Denmark (1044-00158A), the Novo Nordisk Synergy grant (NNF200C0063709), VILLUM FONDEN (40578), the Pioneer Centre for AI (DNRF grant number P1), and the Novo Nordisk Foundation through the MLSS Center (Basic Machine Learning Research in Life Science, NNF20OC0062606).

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

# Appendix

## A License and code availability

The codebase is publicly available at `https://github.com/petergroth/kermut` under the open source MIT License.

## B GP details

### B.1 Structure kernel

The structure kernel is comprised of three components, each increasing model flexibility. The site-comparison kernel, $k_H$, compares site-specific, structure-conditioned amino acid distributions. Given two such discrete probability distributions, $p := f_{\text{IF}}(\mathbf{x})$ and $q := f_{\text{IF}}(\mathbf{x}')$, their distance is quantified via the Hellinger distance [75]

$$d_H(p, q) = \frac{1}{\sqrt{2}} \sqrt{\sum_{i=1}^{20} \left( \sqrt{p_i} - \sqrt{q_i} \right)^2},$$

which is used in the Hellinger kernel [66]

$$k_H(\mathbf{x}, \mathbf{x}') = \exp\left(-\gamma_1 d_H(p, q)\right) = \exp\left( -\gamma_1 \frac{1}{\sqrt{2}} \sqrt{\sum_{i=1}^{20} \left( \sqrt{p_i} - \sqrt{q_i} \right)^2} \right).$$

For the mutation probability kernel, $k_p$, we use the log-probabilities rather than the amino acid identities to reflect that amino acids with similar probability on sites with similar distributions should have similar biochemical effects on the protein. We do not include the log-probabilities of the wild type amino acids, as the inverse folding model by definition is trained to assign high probabilities to the wild type sequence. The exact probability of a wild type amino acid depends on how many other amino acids that are likely to be at the given site. For example, we would expect a probability close to one for a functionally critical amino acid at a particular site. Conversely, for a less critical surface-level residue requiring, e.g., a polar uncharged amino acid, we would expect similar probabilities for the four amino acids of this type. Thus, variations of log-probabilities of the wild type amino acids should be reflected in the distribution on the sites captured by the Hellinger kernel.

For the per-residue amino acid distributions in $k_H$ and $k_p$, we use ProteinMPNN [52]. ProteinMPNN relies on a random decoding order and thus benefits from multiple samples. We decode the wild type amino acid sequence a total of ten times while conditioning on the full structure and the sequence that has been decoded thus far. We then compute a per-residue average distribution. We use the `v_48_020` weights.

For the distance kernel, $k_d$, we calculate the Euclidean distance between $\alpha$ carbon atoms, where the unit is in Ångstrøm. All wild type structures used in $k_d$ and by ProteinMPNN are predicted via AlphaFold2 [64] and are provided by ProteinGym [7].

The sum over all pairs of mutations in equation 2 is motivated by [63] who showed that a linear model was sufficient to effectively model the mutation effects. Note that $\text{Cov}(Y_1 + Y_2, Y_3 + Y_4) = \text{Cov}(Y_1, Y_3) + \text{Cov}(Y_1, Y_4) + \text{Cov}(Y_2, Y_3) + \text{Cov}(Y_2, Y_4)$. Hence, if the $Y_i$'s are the effects of single mutations, we can find the covariance between two double mutant variants by calculating the pairwise sum and similarly for other number of mutations.

### B.2 Sequence kernel

For the sequence kernel, we use embeddings extracted from the ESM-2 protein language model [16]. We use the `esm2_t36_650M_UR50D` model with 650M parameters. The embeddings are mean-pooled across the sequence dimension such that each variant is represented by a 1280 dimensional vector. While it has been shown that other aggregation method can lead to large increases in performance [76], we consider alternate methods such as training a bottleneck model out of the scope of this paper.

### B.3 Parametrization note

An alternative formulation of the kernel, where we omit $\lambda$ from Equation (1) and $\pi$ from Equation (4), and instead provide the structure and sequence kernels with separate coefficients have shown to provide close to identical results when using a smoothed box prior (constrained between 0 and 1). While this approach is somewhat more elegant, this does not justify the re-computation of all results.

### B.4 Zero-shot mean function

For the zero-shot mean function, we download and use the pre-computed zero-shot scores from ProteinGym at `https://github.com/OATML-Markslab/ProteinGym`. The zero-shot value is calculated as the log-likelihood ratio between the variant and the wild type at the mutated residue as described in [10]:

$$f_0(\mathbf{x}) = \sum_{i \in M} \log p(\mathbf{x}_i) - \log p(\mathbf{x}_i^{\text{WT}})$$

The values can be computed straightforwardly using the ESM suite (using the `masked-marginals` strategy). For multi-mutants, the sum of the ratios is taken.

### B.5 Kernel proof

We will in the following argue why Kermut's structure kernel is a valid kernel. Recall that a function $k : \mathcal{X} \times \mathcal{X} \to \mathbb{R}$ is a kernel if and only if the matrix $K$, where $K_{ij} = k(x_i, x_j)$, is symmetric positive semi-definite [65]. From literature we know a number of kernels and certain ways these can be combined to create new kernels. We will argue that Kermut is a kernel by showing how it is composed of known kernels, combined using valid methods.

Note that, if we have a mapping $f : \mathcal{X} \to \mathcal{Z}$ and a kernel $k_Z$, on $\mathcal{Z}$, then $k_X$ defined by $k_X(x, x') := k_Z(f(x), f(x'))$ is a kernel on $\mathcal{X}$.

Let $\mathcal{X}$ be the space of sequences parameterized with respect to a reference sequence. Let $f_1 : \mathcal{X} \to \mathcal{Z}$ be a transformation of the sequences defined as in Section 3.3. $k_{\text{seq}}$ is the squared exponential kernel on the transformed variants. Hence, $k_{\text{seq}}$ is is a kernel on $\mathcal{X}$.

Let $\mathcal{X}^1 \subset \mathcal{X}$ denote the subspace of single mutant variants and $f_2 : \mathcal{X}^1 \to \mathbb{R}^3$ be a function mapping single mutant variants into the 3D coordinates of the $\alpha$-carbon of the particular mutation. $k_d$ is the exponential kernel on this transformed space. Thus $k_d$ is a kernel on $\mathcal{X}^1$.

Let $f_{\text{IF}} : \mathcal{X}^1 \to \mathcal{G} \subseteq [0,1]^{20}$ be defined as in Section 3.3, where $\mathcal{G}$ is the space of probability distributions over the 20 amino acids. $f_{\text{IF}}(\mathbf{x})$ is the probability distribution over the mutated site of $\mathbf{x}$ given by an inverse folding model. $k_H$ is the Hellinger kernel on the single mutation variants transformed by $f_{\text{IF}}$, hence, a valid kernel on $\mathcal{X}^1$. Likewise $k_p$ is the exponential kernel of a transformation, $f_{\text{IF}_1} : \mathcal{X}^1 \to [0,1]$ as defined in Section 3.3, of the sequences, hence also a kernel.

Scaling, multiplying, and adding kernels result in new kernels, making $k_{\text{struct}}^1$ a valid kernel for single mutations [65]. We need to show that $k_{\text{struct}}$ and thereby $k$ is valid for any number of mutations.

Let $f_4 : \mathcal{X} \to \mathcal{B}$ be a function taking a variant $\mathbf{x}$ with $M$ mutations and mapping it to a set $b = \{\mathbf{x}^m\}_{m \in M}$ of all the single mutations which constitutes $\mathbf{x}$. Define the set kernel [77]

$$k_{\text{set}}(b, b') := \sum_{\mathbf{x}^m \in b, \mathbf{x}'^m \in b'} \lambda k_{\text{struct}}^1(\mathbf{x}^m, \mathbf{x}'^m)$$

$k_{\text{struct}}$ is the set kernel on the transformed input and thus also a kernel. We have thereby shown that Kermut is a kernel for variants with any number of mutations.

### B.6 Automatic Model Selection

Kermut has been developed from the ground up as described Section 3.3, which led to separate structure and sequence kernels which are added together. Alternatively, an automatic model selection scheme can be employed for data-driven kernel composition as proposed in Chapter 3 in [78]. For demonstrative purposes, we conduct such a model selection procedure. We define four base kernels

($k_H$, $k_p$, $k_d$, $k_{seq}$) as well as sum and product operations. We choose a subset of 17 ProteinGym assays and fit a GP (with a zero-shot mean function) and each of the four kernels, where the best performing kernel across splits is kept. For the second round, the remaining three kernels are either added or multiplied to the existing kernel. This process is continued until either all four kernels are used or until the test performance no longer increases. The results can be seen in Table B.1, where the final kernel is the product of the four base kernels. The results on the 174 ablation datasets using this kernel (denoted as "Kermut (product)") can be seen with the main Kermut GP in Table F.2. The ECE/ENCE values for Kermut (product) are 0.060/0.192 vs. 0.051/0.170 for the main formulation. Due to the potential for interpretability of the sum-formulation as well as slightly better calibration, the main formulation of the Kermut GP remains as in Equation (4).

Table B.1: Automatic Model Selection with four base kernels and sum/product operations. The resulting kernel is the product of all base kernels.

| Kernel | Round 1 | Round 2 | Round 3 | Round 4 |
|---|---|---|---|---|
| $k_{seq}$ | 0.555 | | | |
| $k_d$ | **0.589** | | | |
| $k_H$ | 0.584 | | | |
| $k_p$ | 0.521 | | | |
| $k_d \times k_{seq}$ | | **0.653** | | |
| $k_d + k_{seq}$ | | 0.642 | | |
| $k_d \times k_H$ | | 0.623 | | |
| $k_d + k_H$ | | 0.622 | | |
| $k_d \times k_p$ | | 0.633 | | |
| $k_d + k_p$ | | 0.629 | | |
| $k_d \times k_{seq} \times k_H$ | | | 0.666 | |
| $k_d \times k_{seq} + k_H$ | | | 0.657 | |
| $k_d \times k_{seq} \times k_p$ | | | **0.678** | |
| $k_d \times k_{seq} + k_p$ | | | 0.666 | |
| $k_d \times k_{seq} \times k_p \times k_H$ | | | | **0.684** |
| $k_d \times k_{seq} \times k_p + k_H$ | | | | 0.676 |

## C    Implementation details

### C.1    General details

We build our kernel using the GPyTorch framework [79]. We assume a homoschedastic Gaussian noise model, on which we place a HalfCauchy prior [80] with scale 0.1. We fit the hyperparameters by maximizing the exact marginal likelihood with gradient descent using the AdamW optimizer [81] with learning rate 0.1 for a 150 steps, which proved to be sufficient for convergence for a number of sampled datasets.

### C.2    System details

All experiments are performed on a Linux-based cluster running Ubuntu 20.04.4 LTS, with a AMD EPYC 7642 48-Core Processor with 192 threads and 1TB RAM. NVIDIA A40s were used for GPU acceleration both for fitting the Gaussian processes and for generating the protein embeddings.

### C.3    Compute time

There are multiple factors to consider when evaluating the training time of Kermut. A limitation of using a Gaussian process framework is the cubic scaling with dataset size. For this reason, we ran the ablation study on only 174 of the 217 datasets.

Training Kermut on these datasets for a single split-scheme using the aforementioned hardware (single GPU) takes approximately 1 hour and 30 minutes. Getting ablation results for all three schemes thus takes between 4-5 hours. This however assumes that

1. the embeddings for the sequence kernel have been precomputed,
2. the probability distribution for all sites in the wild type protein have been precomputed,
3. and that the zero-shot scores have been precomputed.

We do not see any of these as major limitations as the same applies to ProteinNPT and similar models.

Scaling the experiments to the full ProteinGym benchmark is however costly. We were able to train/evaluate Kermut on 215/217 datasets using an NVIDIA A40 GPU with 48GB VRAM. The remaining two, `POLG_CXB3N_Mattenberger_2021` and `POLG_DEN26_Suphatrakul_2023`, datasets were too large to fit into GPU memory without resorting to reduced precision. For these, we trained and evaluated the model using CPU only which takes considerable time.

## C.4  Handling long sequences

The structures used for obtaining the site-wise probability distributions are predicted by AlphaFold2 [64] and are provided in the ProteinGym repository [7]. The provided structures for `A0A140D2T1_ZIKV` and `POLG_HCVJF` do not contain the full structures however, but only a localized area where the mutations occur due to the long sequence lengths. Since our model only operates on sites with mutations, this is not an issue for neither the inverse-folding probability distributions nor the inter-residue distances.

For `BRCA2_HUMAN`, three PDB files are provided due to the long wild type sequence length. We use ProteinMPNN to obtain the distributions at all sites in each PDB file and stitch them together in a preprocessing step. Calculating the inter-residue distances is however non-trivial and would require a careful alignment of the three structures. Instead, we drop the distance term, $k_d$, in the kernel for the `BRCA2_HUMAN_Erwood_2022_HEK293T` dataset (equivalently setting it to one: $k_d(\mathbf{x}, \mathbf{x}') = 1$).

The sequence kernel operates on ESM-2 embeddings. The ESM-2 model has a maximum sequence length of 1022 amino acids. Protein sequences that are longer than this limit are truncated.

### C.4.1  Example wall-clock time

The wall-clock times for generating the results used for the calibration curves in Figures 3a and 3b and Appendix L.2 for one split scheme can be seen in Table C.1 for Kermut and ProteinNPT. The experiments were carried out using identical hardware. The test system is however a shared compute cluster with sharded GPUs, so variance is expected between runs. Generating the full results for the figures thus takes Kermut approximately 10 minutes while ProteinNPT takes 400 hours. This shows the significant reduction in computational burden that Kermut allows for. Both ProteinNPT and Kermut assumes that sequence embeddings are available a priori.

Table C.1: Approximate wall clock times for training and evaluating Kermut and ProteinNPT for a single split scheme, i.e., by using 5-fold cross validation. While the runtime of Kermut scales with dataset size, ProteinNPT appears to scale more strongly with sequence length due to the tri-axial attention.

| Dataset | Kermut runtime | PNPT runtime | $N$ | $L$ |
|---|---|---|---|---|
| BLAT_ECOLX | 111s | $\approx 32h$ | 4996 | 286 |
| PA_I34A1 | 45s | $\approx 52h$ | 1820 | 716 |
| TCRG1_MOUSE | 19s | $\approx 22h$ | 621 | 37 |
| OPSD_HUMAN | 14s | $\approx 40h$ | 165 | 348 |

## C.5  Computational complexity

Evaluating the kernel for two variants $\mathbf{x}$ and $\mathbf{x}'$, involves computation of the two components $k_{\text{seq}}$ and $k_{\text{struct}}$:

For each variant $k_{\text{seq}}$ requires a forward pass through ESM-2, which is based on the transformer architecture and has quadratic scaling with respect to the sequence length. Given the ESM-2 embeddings, the computational complexity is constant in sequence length and number of mutations.

$k_{\text{struct}}$ requires a single forward pass through ProteinMPNN for the wild type protein. Given the output of ProteinPMNN, the computational complexity for evaluation of the kernel is $m_1 \times m_2$ for two proteins with $m_1$ and $m_2$ number of mutations. The computational complexity of $k_{\text{struct}}^1$ is constant in sequence length and number of mutations.

# D   Data

All data and evaluation software is accessed via the ProteinGym [7] repository at `https://github.com/OATML-Markslab/ProteinGym` which is under the MIT License.

# E   Detailed results

The ProteinGym suite provides an aggregation procedure, whereby the predictive performance across both cross-validation schemes and functional categories can be gauged. We provide these results in Tables E.1 to E.4. We mirror the label normalization from [19] and [7], where, for each fold of cross-validation, train and test labels are normalized given the mean and standard deviation of the training labels. The Spearman correlation coefficient and MSE are then computed in the normalized space across folds, leading to a single Spearman coefficient and MSE per assay per split (i.e., not as an average of metrics per CV-fold).

Table E.1: Aggregated Spearman results on the ProteinGym substitution benchmark. Performance is shown per cross-validation scheme. Kermut reaches superior performance across the board. The fifth data column shows the non-parametric bootstrap standard error of the difference between the Spearman performance for each model and Kermut, computed over 10,000 bootstrap samples from the set of proteins in the ProteinGym substitution benchmark.

| Model name | Spearman per scheme (↑) | | | | Std. err. |
| | Cont. | Mod. | Rand. | Avg. | |
| --- | --- | --- | --- | --- | --- |
| Kermut | **0.610** | **0.633** | **0.744** | **0.662** | 0.000 |
| ProteinNPT | 0.547 | 0.564 | 0.730 | 0.613 | 0.009 |
| Tranception Emb. | 0.490 | 0.526 | 0.696 | 0.571 | 0.008 |
| MSAT Emb. | 0.525 | 0.538 | 0.642 | 0.568 | 0.013 |
| ESM-1v Emb. | 0.481 | 0.506 | 0.639 | 0.542 | 0.011 |
| TranceptEVE + OHE | 0.441 | 0.440 | 0.550 | 0.477 | 0.012 |
| Tranception + OHE | 0.419 | 0.419 | 0.535 | 0.458 | 0.012 |
| MSAT + OHE | 0.410 | 0.412 | 0.536 | 0.453 | 0.014 |
| DeepSequence + OHE | 0.400 | 0.400 | 0.521 | 0.440 | 0.016 |
| ESM-1v + OHE | 0.367 | 0.368 | 0.514 | 0.417 | 0.014 |
| OHE | 0.064 | 0.027 | 0.579 | 0.224 | 0.014 |

Table E.2: Aggregated Spearman results on the ProteinGym substitution benchmark. Performance is shown per functional category. Kermut reaches superior performance across the board.

| Model name | Spearman per function (↑) | | | | |
| | Activity | Binding | Expression | Fitness | Stability |
| --- | --- | --- | --- | --- | --- |
| Kermut | **0.606** | **0.630** | **0.672** | **0.581** | **0.824** |
| ProteinNPT | 0.577 | 0.536 | 0.637 | 0.545 | 0.772 |
| Tranception Emb. | 0.520 | 0.529 | 0.613 | 0.519 | 0.674 |
| MSAT Emb. | 0.547 | 0.470 | 0.584 | 0.493 | 0.749 |
| ESM-1v Emb. | 0.487 | 0.450 | 0.587 | 0.468 | 0.717 |
| TranceptEVE + OHE | 0.502 | 0.444 | 0.476 | 0.470 | 0.493 |
| Tranception + OHE | 0.475 | 0.416 | 0.476 | 0.448 | 0.473 |
| MSAT + OHE | 0.480 | 0.393 | 0.463 | 0.437 | 0.491 |
| DeepSequence + OHE | 0.467 | 0.418 | 0.424 | 0.422 | 0.471 |
| ESM-1v + OHE | 0.421 | 0.363 | 0.452 | 0.383 | 0.463 |
| OHE | 0.213 | 0.212 | 0.226 | 0.194 | 0.273 |

Table E.3: Aggregated MSE results on the ProteinGym substitution benchmark. Performance is shown per cross-validation scheme. Kermut reaches superior performance across the board. The fifth data column shows the non-parametric bootstrap standard error of the difference between the MSE performance for each model and Kermut, computed over 10,000 bootstrap samples from the set of proteins in the ProteinGym substitution benchmark.

| Model name | MSE per scheme (↓) | | | | Std. err. |
| | Cont. | Mod. | Rand. | Avg. | |
|---|---|---|---|---|---|
| Kermut | **0.699** | **0.652** | **0.414** | **0.589** | 0.000 |
| ProteinNPT | 0.820 | 0.771 | 0.459 | 0.683 | 0.017 |
| MSAT Emb. | 0.836 | 0.795 | 0.573 | 0.735 | 0.021 |
| Tranception Emb. | 0.972 | 0.833 | 0.503 | 0.769 | 0.023 |
| ESM-1v Emb. | 0.937 | 0.861 | 0.563 | 0.787 | 0.030 |
| TranceptEVE + OHE | 0.953 | 0.914 | 0.743 | 0.870 | 0.019 |
| MSAT + OHE | 0.963 | 0.934 | 0.749 | 0.882 | 0.020 |
| DeepSequence + OHE | 0.967 | 0.940 | 0.767 | 0.891 | 0.017 |
| Tranception + OHE | 0.985 | 0.934 | 0.766 | 0.895 | 0.022 |
| ESM-1v + OHE | 0.977 | 0.949 | 0.764 | 0.897 | 0.013 |
| OHE | 1.158 | 1.125 | 0.898 | 1.061 | 0.017 |

Table E.4: Aggregated results on the ProteinGym substitution benchmark. Performance is shown per functional category. Kermut reaches superior performance across the board.

| Model name | MSE per function (↓) | | | | |
| | Activity | Binding | Expression | Fitness | Stability |
|---|---|---|---|---|---|
| Kermut | **0.630** | **0.843** | **0.523** | **0.657** | **0.289** |
| ProteinNPT | 0.703 | 1.016 | 0.578 | 0.752 | 0.368 |
| MSAT Emb. | 0.728 | 1.092 | 0.660 | 0.789 | 0.405 |
| Tranception Emb. | 0.814 | 1.080 | 0.639 | 0.788 | 0.525 |
| ESM-1v Emb. | 0.799 | 1.231 | 0.655 | 0.792 | 0.456 |
| TranceptEVE + OHE | 0.793 | 1.199 | 0.780 | 0.825 | 0.756 |
| MSAT + OHE | 0.810 | 1.221 | 0.788 | 0.836 | 0.756 |
| DeepSequence + OHE | 0.830 | 1.140 | 0.832 | 0.860 | 0.793 |
| Tranception + OHE | 0.831 | 1.246 | 0.787 | 0.845 | 0.765 |
| ESM-1v + OHE | 0.843 | 1.192 | 0.795 | 0.870 | 0.783 |
| OHE | 1.022 | 1.306 | 0.986 | 1.040 | 0.949 |

## E.1 Results on new splits

The modulo and contiguous splits in ProteinGym were updated in April 2024. For completeness, we here provide results using Kermut on the updated splits for all 217 DMS assays. These can be seen in Table E.5, where we observe a slight decrease in performance for the contiguous and modulo splits, compared to the reference results in Table 1.

Table E.5: Performance on the ProteinGym benchmark using the corrected splits.

| Model type | Model name | Spearman (↑) | | | | MSE (↓) | | | |
| | | Contig. | Mod. | Rand. | Avg. | Contig. | Mod. | Rand. | Avg. |
|---|---|---|---|---|---|---|---|---|---|
| GP | Kermut | 0.591 | 0.631 | 0.744 | 0.655 | 0.739 | 0.680 | 0.141 | 0.611 |

# F Ablation results

In Section 4.1, an ablation study was carried out by removing components of Kermut. In Table 2, the performance difference in Spearman correlation was shown. In Table F.1 we see the performance difference in MSE. The aggregated absolute Spearman and MSE values are shown in Tables F.2 and F.3, while we show the performance per functional category in Tables F.4 and F.5. These results suggest that an even wider combinatorial examination of Kermut's kernel composition might lead to slightly increased performance, e.g., by multiplying a Matérn 5/2 sequence kernel with the structure kernel. We must however note that the standard errors in Tables F.2 and F.3 suggest that while the product and Matérn configurations lead to better results on the ablation datasets, the differences are not significant.

In addition to the shown methods, we considered sequence-only kernels as a baselines. One example is the inverse-multiquadratic Hamming (IMQ-H) kernel from [31]. This kernel, however, relies on the Hamming distance between one-hot encoded sequences for its covariances. For ProteinGym's single-mutant benchmark, this is not sufficient as the Hamming distance between all variants is 2, resulting in constant predictions for all folds and subsequent Spearman correlations narrowly centered on 0.

Table F.1: Ablation results. Key components of the kernel are removed and the model is trained and evaluated on 174/217 assays from the ProteinGym benchmark. The ablation column shows the alteration to the GP formulation. The metrics are subtracted from Kermut to show the change in performance. Positive $\Delta$MSE values indicate drop in performance.

| Description | Ablation | $\Delta$MSE | | | |
| --- | --- | --- | --- | --- | --- |
| | | Contig. | Mod. | Rand. | Avg. |
| No structure kernel | $k_{\text{struct}} = 0$ | 0.096 | 0.087 | 0.046 | 0.076 |
| No sequence kernel | $k_{\text{seq}} = 0$ | 0.060 | 0.059 | 0.077 | 0.065 |
| No inter-residue dist. | $k_d = 1$ | 0.058 | 0.060 | 0.011 | 0.043 |
| No mut. prob./site comp. | $k_p = k_H = 1$ | 0.046 | 0.040 | 0.024 | 0.036 |
| Const. mean | $m(\mathbf{x}) = \alpha$ | 0.042 | 0.036 | 0.015 | 0.030 |
| No mut. prob. | $k_p = 1$ | 0.032 | 0.021 | 0.022 | 0.024 |
| No site comp. | $k_H = 1$ | 0.005 | 0.009 | 0.008 | 0.006 |

Table F.2: Ablation results. Key components of the kernel are removed or altered and the model is trained and evaluated on 174/217 assays from the ProteinGym benchmark. The ablation column shows the alteration to the kernel formulation.

| Description | Ablation | Spearman ($\uparrow$) | | | | Std. err. |
| --- | --- | --- | --- | --- | --- | --- |
| | | Contig. | Mod. | Rand. | Avg. | |
| No structure kernel | $k_{\text{struct}} = 0$ | 0.523 | 0.550 | 0.710 | 0.594 | 0.009 |
| No sequence kernel | $k_{\text{seq}} = 0$ | 0.565 | 0.582 | 0.695 | 0.614 | 0.006 |
| No inter-residue dist. | $k_d = 1$ | 0.556 | 0.577 | 0.739 | 0.624 | 0.004 |
| No mut. prob./site comp. | $k_p = k_H = 1$ | 0.570 | 0.591 | 0.737 | 0.633 | 0.006 |
| Const. mean | $m(\mathbf{x}) = \alpha$ | 0.569 | 0.596 | 0.735 | 0.633 | 0.007 |
| No mut. prob. | $k_p = 1$ | 0.583 | 0.609 | 0.738 | 0.643 | 0.004 |
| No site comp. | $k_H = 1$ | 0.599 | 0.619 | 0.743 | 0.653 | 0.002 |
| Kermut (SE in $k_H$) | $k_H = k_{\text{SE}}$ | 0.598 | 0.621 | 0.745 | 0.655 | 0.002 |
| Kermut (JSD in $k_H$) | $k_H = k_{\text{JSD}}$ | 0.604 | 0.626 | 0.745 | 0.658 | 0.001 |
| Kermut (Matérn in $k_{\text{seq}}$) | $k_{\text{seq}} = k_{\text{Matérn5/2}}$ | **0.608** | 0.629 | 0.746 | 0.661 | 0.002 |
| Kermut (product) | $k = k_{\text{struct}} \times k_{\text{seq}}$ | 0.606 | **0.632** | **0.750** | **0.662** | 0.003 |
| Kermut | | 0.605 | 0.628 | 0.743 | 0.659 | 0.000 |

Table F.3: Ablation results. Key components of the kernel are removed or altered and the model is trained and evaluated on 174/217 assays from the ProteinGym benchmark. The ablation column shows the alteration to the kernel formulation.

| Description | Ablation | MSE (↓) | | | | Std. err. |
|---|---|---|---|---|---|---|
| | | Contig. | Mod. | Rand. | Avg. | |
| No structure kernel | $k_{\text{struct}} = 0$ | 0.825 | 0.769 | 0.460 | 0.684 | 0.012 |
| No sequence kernel | $k_{\text{seq}} = 0$ | 0.791 | 0.744 | 0.492 | 0.676 | 0.008 |
| No inter-residue dist. | $k_d = 1$ | 0.789 | 0.743 | 0.426 | 0.652 | 0.006 |
| No mut. prob./site comp. | $k_p = k_H = 1$ | 0.775 | 0.722 | 0.436 | 0.644 | 0.007 |
| Const. mean | $m(\mathbf{x}) = \alpha$ | 0.770 | 0.718 | 0.429 | 0.639 | 0.005 |
| No mut. prob. | $k_p = 1$ | 0.761 | 0.704 | 0.436 | 0.634 | 0.006 |
| No site comp. | $k_H = 1$ | 0.735 | 0.691 | 0.421 | 0.616 | 0.002 |
| Kermut (SE in $k_H$) | $k_H = k_{\text{SE}}$ | 0.738 | 0.690 | 0.416 | 0.614 | 0.003 |
| Kermut (JSD in $k_H$) | $k_H = k_{\text{JSD}}$ | 0.731 | 0.684 | 0.415 | 0.610 | 0.002 |
| Kermut (Matérn in $k_{\text{seq}}$) | $k_{\text{seq}} = k_{\text{Matérn5/2}}$ | **0.724** | **0.678** | 0.414 | **0.605** | 0.002 |
| Kermut (product) | $k = k_{\text{struct}} \times k_{\text{seq}}$ | 0.726 | **0.678** | **0.411** | **0.605** | 0.004 |
| Kermut | | 0.730 | 0.683 | 0.420 | 0.611 | 0.000 |

Table F.4: Ablation results. Key components of the kernel are removed or altered and the model is trained and evaluated on 174/217 assays from the ProteinGym benchmark. The ablation column shows the alteration to the kernel formulation. Performance is shown per functional category.

| Model name | Ablation | Spearman per function (↑) | | | | |
|---|---|---|---|---|---|---|
| | | Activity | Binding | Expression | Fitness | Stability |
| Const. mean | $m(\mathbf{x}) = \alpha$ | 0.579 | 0.580 | 0.651 | 0.536 | 0.821 |
| No site comp. | $k_H = 1$ | 0.590 | 0.619 | 0.664 | 0.574 | 0.820 |
| No mut. prob. | $k_p = 1$ | 0.590 | 0.611 | 0.651 | 0.564 | 0.801 |
| No mut. prob./site comp. | $k_p = k_H = 1$ | 0.569 | 0.601 | 0.648 | 0.557 | 0.790 |
| No inter-residue dist. | $k_d = 1$ | 0.562 | 0.560 | 0.634 | 0.546 | 0.818 |
| No sequence kernel | $k_{\text{seq}} = 0$ | 0.578 | 0.616 | 0.611 | 0.547 | 0.716 |
| No structure kernel | $k_{\text{struct}} = 0$ | 0.531 | 0.529 | 0.614 | 0.519 | 0.778 |
| Kermut (Matérn in $k_{\text{seq}}$) | $k_{\text{seq}} = k_{\text{Matérn5/2}}$ | **0.604** | 0.625 | 0.667 | **0.581** | 0.828 |
| Kermut (SE in $k_H$) | $k_H = k_{\text{SE}}$ | 0.593 | 0.623 | 0.664 | 0.576 | 0.818 |
| Kermut (JSD in $k_H$) | $k_H = k_{\text{JSD}}$ | 0.599 | 0.628 | 0.664 | 0.578 | 0.823 |
| Kermut (product) | $k = k_{\text{struct}} \times k_{\text{seq}}$ | 0.599 | **0.632** | **0.674** | 0.578 | **0.829** |
| Kermut | | 0.602 | 0.625 | 0.665 | 0.578 | 0.824 |

Table F.5: Ablation results. Key components of the kernel are removed or altered and the model is trained and evaluated on 174/217 assays from the ProteinGym benchmark. The ablation column shows the alteration to the kernel formulation. Performance is shown per functional category.

| Model name | Ablation | MSE per function (↓) | | | | |
|---|---|---|---|---|---|---|
| | | Activity | Binding | Expression | Fitness | Stability |
| Const. mean | $m(\mathbf{x}) = \alpha$ | 0.672 | 0.932 | 0.575 | 0.710 | 0.304 |
| No site comp. | $k_H = 1$ | 0.651 | 0.906 | 0.557 | 0.669 | 0.295 |
| No mut. prob. | $k_p = 1$ | 0.651 | 0.923 | 0.584 | 0.682 | 0.327 |
| No mut. prob./site comp. | $k_p = k_H = 1$ | 0.672 | 0.933 | 0.586 | 0.688 | 0.343 |
| No inter-residue dist. | $k_d = 1$ | 0.696 | 0.965 | 0.599 | 0.702 | 0.301 |
| No sequence kernel | $k_{\text{seq}} = 0$ | 0.673 | 0.926 | 0.622 | 0.718 | 0.440 |
| No structure kernel | $k_{\text{struct}} = 0$ | 0.712 | 0.993 | 0.626 | 0.734 | 0.358 |
| Kermut (Matérn in $k_{\text{seq}}$) | $k_{\text{seq}} = k_{\text{Matérn5/2}}$ | **0.636** | 0.891 | 0.555 | **0.662** | **0.283** |
| Kermut (SE in $k_H$) | $k_H = k_{\text{SE}}$ | 0.648 | 0.899 | 0.558 | 0.669 | 0.299 |
| Kermut (JSD in $k_H$) | $k_H = k_{\text{JSD}}$ | 0.642 | 0.893 | 0.558 | 0.667 | 0.291 |
| Kermut (product) | $k = k_{\text{struct}} \times k_{\text{seq}}$ | 0.643 | **0.884** | **0.548** | 0.667 | **0.283** |
| Kermut | | 0.640 | 0.903 | 0.558 | 0.665 | 0.289 |

# G    Results for multi-mutants in ProteinGym

69 of the datasets from the ProteinGym benchmark include multi-mutants. In addition to the random, modulo, and contiguous split, these also have a `fold_rand_multiples` split. We here show the results for Kermut in this setting. Of the 69 datasets, we select 52 which (due to the cubic scaling of fitting GPs) include fewer than 7500 variant sequences. We additionally ignore the `GCN4_YEAST_Staller_2018` dataset which has a very large number of mutations. This leads to a total of 51 datasets. All results where the training domain is "1M/2M→" are from models trained using the above split. The "1M→" domain results correspond to training the model once on single mutants and evaluating it on double mutants. In addition to Kermut, we include as a baseline results from a GP using the sequence kernel operating on mean-pooled ESM-2 embeddings. This is equivalent to setting the structure kernel to 0, $k_{\text{struct}} = 0$ as in Table 2. For results on the multi-mutant GB1 landscape from FLIP [6], see Appendix H.

Table G.1: Results in multi-mutant setting. Each row corresponds to a different setting of training and evaluation domain. Third row corresponds to the `fold_rand_multiples` split-scheme from ProteinGym. Experiments are carried out on 51 datasets, corresponding to all datasets with multiple mutants with less than 7500 variants in total with the exception of `GCN4_YEAST_Staller_2018`, which has been removed due to its high mutation count. $^*$: Constant mean.

| Domain | Spearman ($\uparrow$) | | | MSE ($\downarrow$) | | |
|---|---|---|---|---|---|---|
| | Kermut | Kermut$^*$ | $k_{\text{seq}}$ | Kermut | Kermut$^*$ | $k_{\text{seq}}$ |
| 1M/2M→1M | **0.910** | 0.908 | 0.879 | **0.139** | 0.143 | 1.154 |
| 1M/2M→2M | **0.895** | **0.895** | 0.873 | **0.103** | **0.103** | 1.044 |
| 1M/2M→1M/2M | **0.938** | 0.937 | 0.913 | **0.116** | 0.118 | 1.092 |
| 1M→2M | 0.650 | **0.660** | 0.648 | 0.805 | 0.580 | **0.506** |

# H    Results on GB1 landscape from the FLIP benchmark

To further investigate Kermut's performance in a multi-mutant setting, we apply it to the GB1 fitness landscape from FLIP [6]. The results can be seen in Table H.1. Kermut's base configuration severely underperforms in the 1-vs-rest split, while reaching similar correlation coefficients in the 2-vs-rest and 3-vs-rest splits. A reason for the initial low score might be the that the accuracy of zero-shot methods at different mutation orders tend to decrease with higher mutation count. Despite this, we still see low performance compared to other models when removing the zero-shot mean function.

The GB1 landscape is comprised of 149,361 mutations at exactly four highly epistatic positions in the GB1 binding domain of Protein G. This is a challenging task for Kermut, whose structural kernel directly compares sites. With only four sites to compare, Kermut fails to accurately model the fitness landscape. Additionally, as described in the main text, the only epistatic modeling in Kermut is via the mean-pooled protein language model embeddings in the sequence kernel, which proves insufficient to capture the interplay of these highly epistatic sites. Further experimentation is required to thoroughly gauge Kermut's performance across diverse multi-mutant assays where the number of mutated residues is variable and epistasis plays a central role.

Table H.1: Performance on FLIP's GB1 landscape. *: Reference results from FLIP. Best and second best scores per split has been highlighted.

| Model | 1-vs-rest | 2-vs-rest | 3-vs-rest | low-vs-high |
|---|---|---|---|---|
| ESM-1b (per AA)* | 0.28 | 0.55 | 0.79 | **0.59** |
| ESM-1b (mean)* | 0.32 | 0.36 | 0.54 | 0.13 |
| ESM-1b (mut mean) * | -0.08 | 0.19 | 0.49 | 0.45 |
| ESM-1v (per AA)* | 0.28 | 0.28 | **0.82** | **0.51** |
| ESM-1v (mean)* | 0.32 | 0.32 | 0.77 | 0.10 |
| ESM-1v (mut mean)* | 0.19 | 0.19 | 0.80 | 0.49 |
| ESM-untrained (per AA)* | 0.06 | 0.06 | 0.48 | 0.23 |
| ESM-untrained (mean)* | 0.05 | 0.05 | 0.46 | 0.10 |
| ESM-untrained (mut mean)* | 0.21 | 0.21 | 0.57 | 0.13 |
| Ridge* | 0.28 | **0.59** | 0.76 | 0.34 |
| CNN* | 0.17 | 0.32 | **0.83** | **0.51** |
| Levenshtein* | 0.17 | 0.16 | -0.04 | -0.10 |
| BLOSUM62* | 0.15 | 0.14 | 0.01 | -0.13 |
| Kermut | -0.14 | 0.52 | 0.77 | 0.35 |
| Kermut (constant mean) | **0.37** | 0.55 | 0.77 | 0.36 |
| Baseline GP | **0.40** | **0.57** | 0.73 | 0.42 |

# I  Histogram over assays for multi-mutant datasets

A histogram over normalized assay values for 51/69 dataset with multi-mutants (total fewer than 7500 sequences) can be seen in Figure I.1. The histograms are colored according to the number of mutations per variant. The assay distribution belong to different modalities depending on the number of mutations present, where double mutations often lead to a loss of fitness.

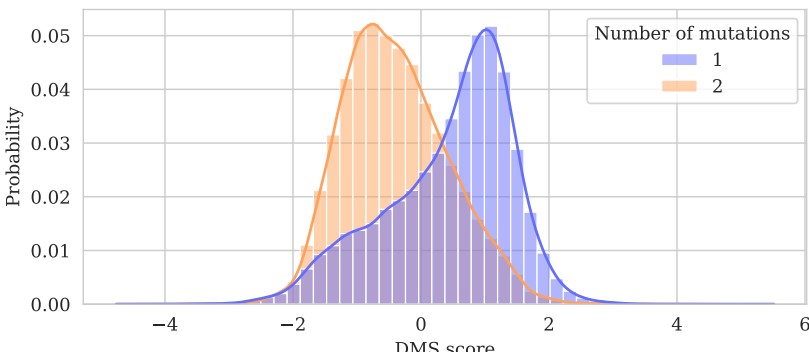

Figure I.1: Histogram over normalized assay values for 51/69 datasets with multi-mutants. All datasets with more than 7500 variants are ignored. The histograms are colored according to the number of mutations per variant. The assay distribution belong to different modalities depending on the number of mutations present, where double mutations commonly lead to a loss of fitness.

# J  Uncertainty calibration

## J.1  Confidence interval-based calibration

Given a collection of mean predictions and uncertainties, we wish to gauge how well-calibrated the uncertainties are. The posterior predictive mean and variance for each data point is interpreted as a Gaussian distribution and symmetric intervals of varying confidence are placed on each prediction [41]. In a well-calibrated model, approximately $x$ % of predictions should lie within a $x$ % confidence

interval, e.g., $50\%$ of observations should fall in the $50\%$ confidence interval. The confidence intervals are discretized into $K$ bins and the fraction of predictions falling within in bin is calculated. The calibration curve then plots the confidence intervals vs. the fractions, whereby a diagonal line corresponds to perfect calibration. Given the fractions and confidence intervals, the *expected calibration error* (ECE) is calculated as

$$\text{ECE} = \frac{1}{K} \sum_{i=1}^{K} |\text{acc}(i) - i|,$$

where $K$ is the number of bins, $i$ indicates the equally spaced confidence intervals, and $\text{acc}(i)$ is the fraction of predictions falling within the $i$th confidence interval.

## J.2   Error-based calibration

An alternative method of gauging calibratedness is error-based calibration where the prediction error is tied directly to predictions [41, 42]. The predictions are sorted according to their predictive uncertainty and placed into $K$ bins. For each bin, the root mean square error (RMSE) and root mean variance (RMV) is computed. In error-based calibration, a well-calibrated model as equal RMSE and RMV, i.e., a diagonal line. The $x$ and $y$ values in the resulting calibration plot are however not normalized from 0 to 1 as in confidence interval-based calibration. The *expected normalized calibration error* (ENCE) can be computed as

$$\text{ENCE} = \frac{1}{K} \sum_{i=1}^{K} \frac{|\text{RMV}(i) - \text{RMSE}(i)|}{\text{RMV}(i)}.$$

Additionally, we compute the *coefficient of variation* $(c_v)$ as

$$c_v = \frac{\sqrt{\frac{\sum_{n=1}^{N}(\sigma_n - \mu_\sigma)^2}{N-1}}}{\mu_\sigma},$$

where $\mu_\sigma = \frac{1}{N} \sum_{n=1}^{N} \sigma_n$, and where $n$ indexes the $N$ data points [42].

## J.3   Uncertainty calibration across all datasets

To quantitatively describe the calibratedness of Kermut across all datasets, we compute the above calibration metrics for all split schemes and folds. We do this for Kermut and a baseline GP using the sequence kernel on ESM-2 embeddings (equivalent to the sequence kernel from Equation (3) with a constant mean). These values can be seen in Figure J.1 for each of the three main split-schemes, the 1M/2M→1M/2M split ("Multiples"), and the 1M→2M split ("Extrapolation"). For the three main splits, we see that the calibratedness measured by the ENCE correlates with performance, where the lowest values are seen in the random setting. Generally, we see that the inclusion of the structure kernel improves not only performance (see Table 2) but also calibration, as the ECE and ENCE values are consistently better for Kermut, with the exception of ENCE in the extrapolation domain. We however see that the sequence kernel (squared exponential) consistently provide predictive variances that themselves vary more, which is generally preferable.

Overall, we can conclude that Kermut appears to be well-calibrated both qualitatively and quantitatively. While the ECE values are generally small and similar, the ENCE values suggest a more nuanced calibration landscape, where we can expect low errors when our model predicts low uncertainties, particularly in the random scheme.

For each dataset, split, and fold, we perform a linear regression to the error-based calibration curves and summarize the slope and intercept. As perfect calibration corresponds to a diagonal line, we want the distribution over the slopes to be centred on one and the distribution over intercepts to be centred on zero. Boxplots for this analysis can be seen in Appendix J.3. As indicated in (a), most of Kermut's calibration curves are well-behaved, while the baseline GP in (b) is generally poorly calibrated.

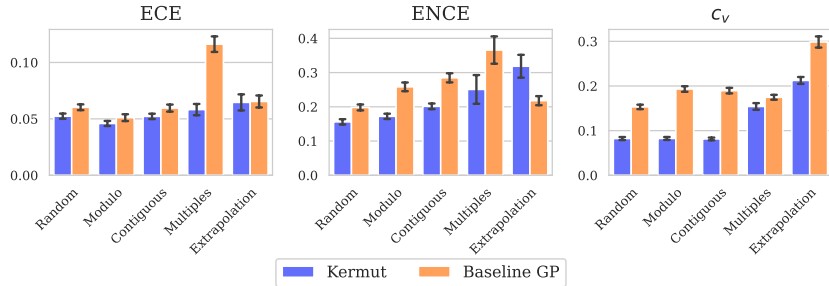

Figure J.1: Calibration metrics per domain for Kermut and the sequence kernel on ESM-2 embeddings. Random, modulo, and contiguous domains are from the ProteinGym substitution benchmark. Multiples corresponds to training and testing on both single and double mutants. Extrapolation corresponds to training on singles and predicting on doubles. 51 datasets with multi-mutants was used for the figure for all domains for comparability. The performance results for the multi-mutant setting can be found in Table G.1. Errorbars correspond to standard error.

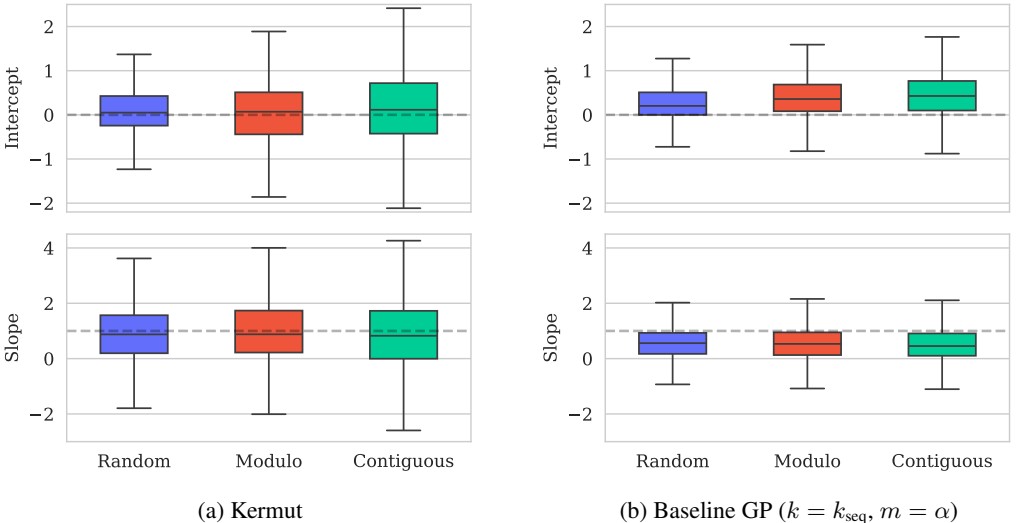

(a) Kermut             (b) Baseline GP ($k = k_{\text{seq}}, m = \alpha$)

Figure J.2: Boxplot of intercepts and slopes of error-based calibration curves for Kermut and a baseline GP with the sequence kernel on ESM-2 embeddings. Perfect calibration has an intercept of zero and a slope of one (indicated by dashed lines). The baseline GP has poor calibration compared to Kermut. Horizontal lines indicate 0.9, 0.75, 0.5, 0.25, 0.1 quantiles.

# K Predicted vs. true values for calibration datasets

## K.1 BLAT_ECOLX_Stiffler_2015

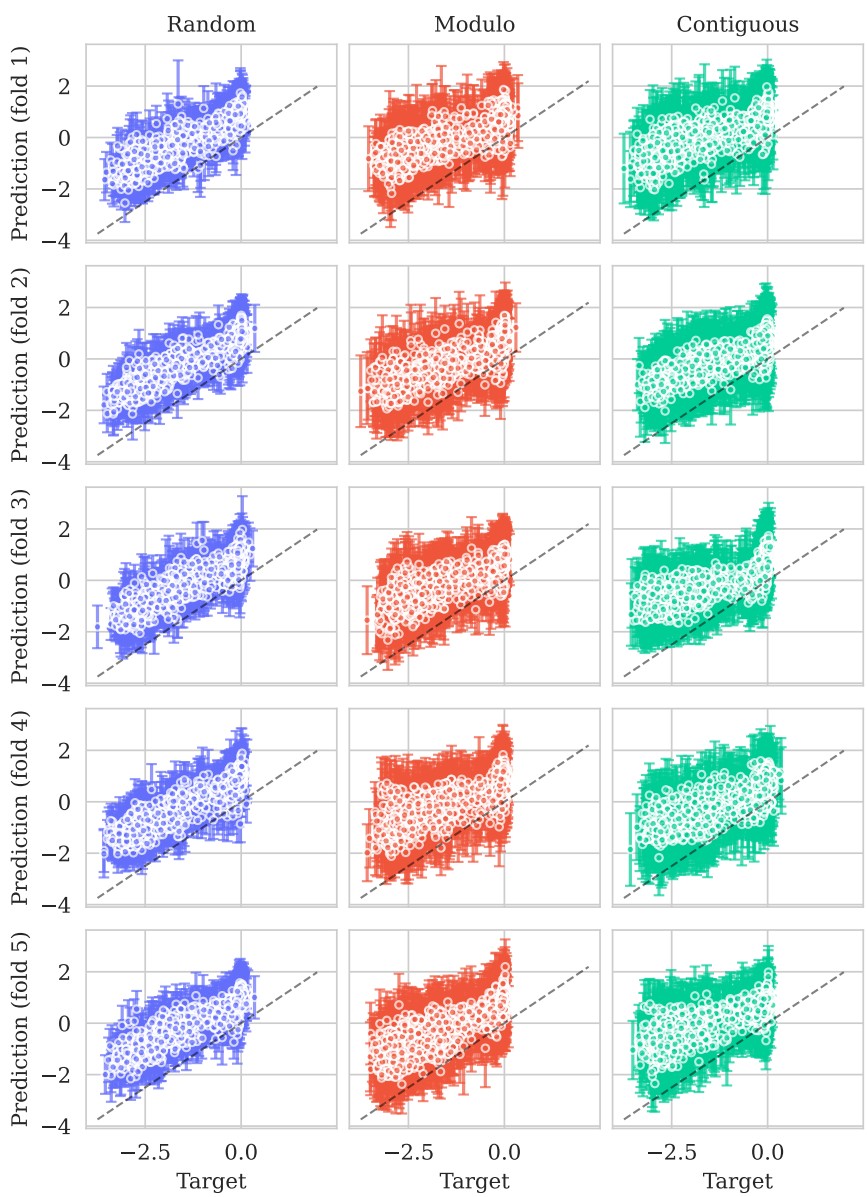

Figure K.1: Predicted means ($\pm 2\sigma$) vs. true values. Columns correspond to CV-schemes. Rows correspond to test folds. Perfect prediction corresponds to dashed diagonal line ($x = y$).

## K.2  PA_I34A1_Wu_2015

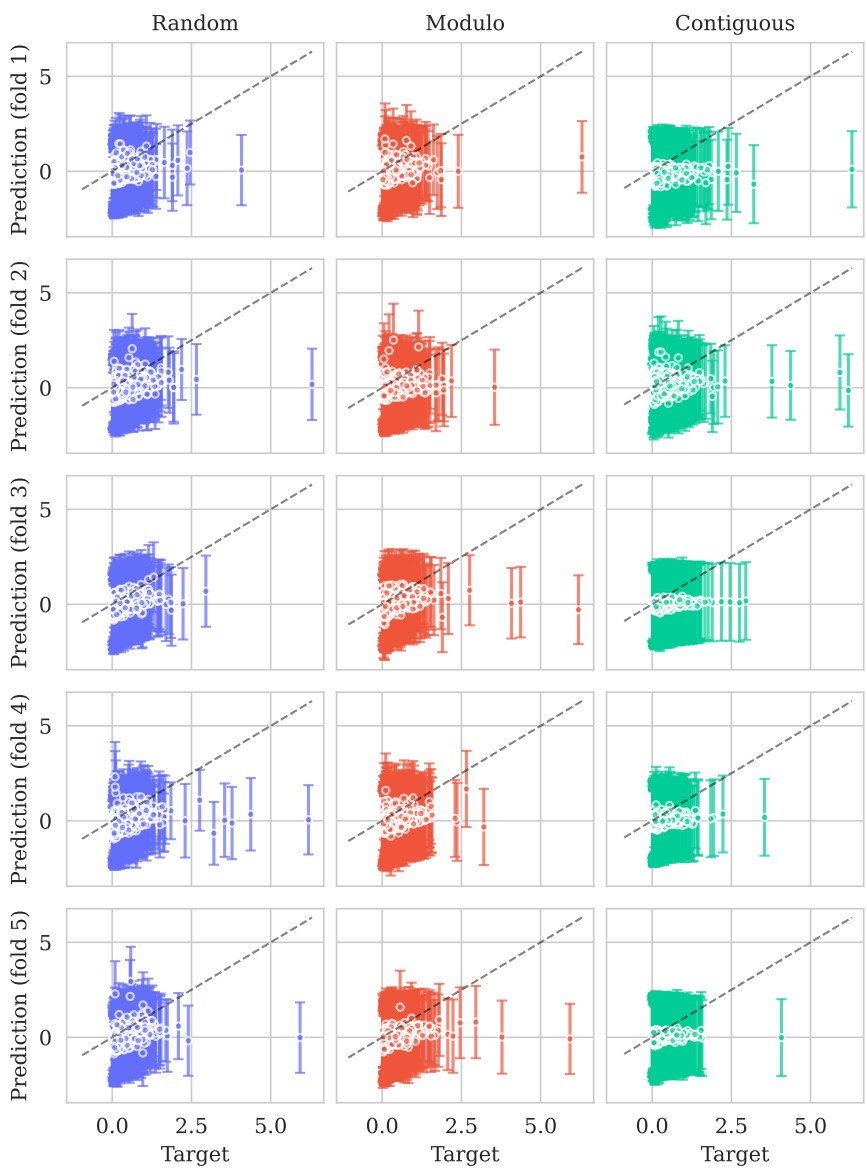

Figure K.2: Predicted means ($\pm 2\sigma$) vs. true values. Columns correspond to CV-schemes. Rows correspond to test folds. Perfect prediction corresponds to dashed diagonal line ($x = y$).

## K.3 TCRG1_MOUSE_Tsuboyama_2023

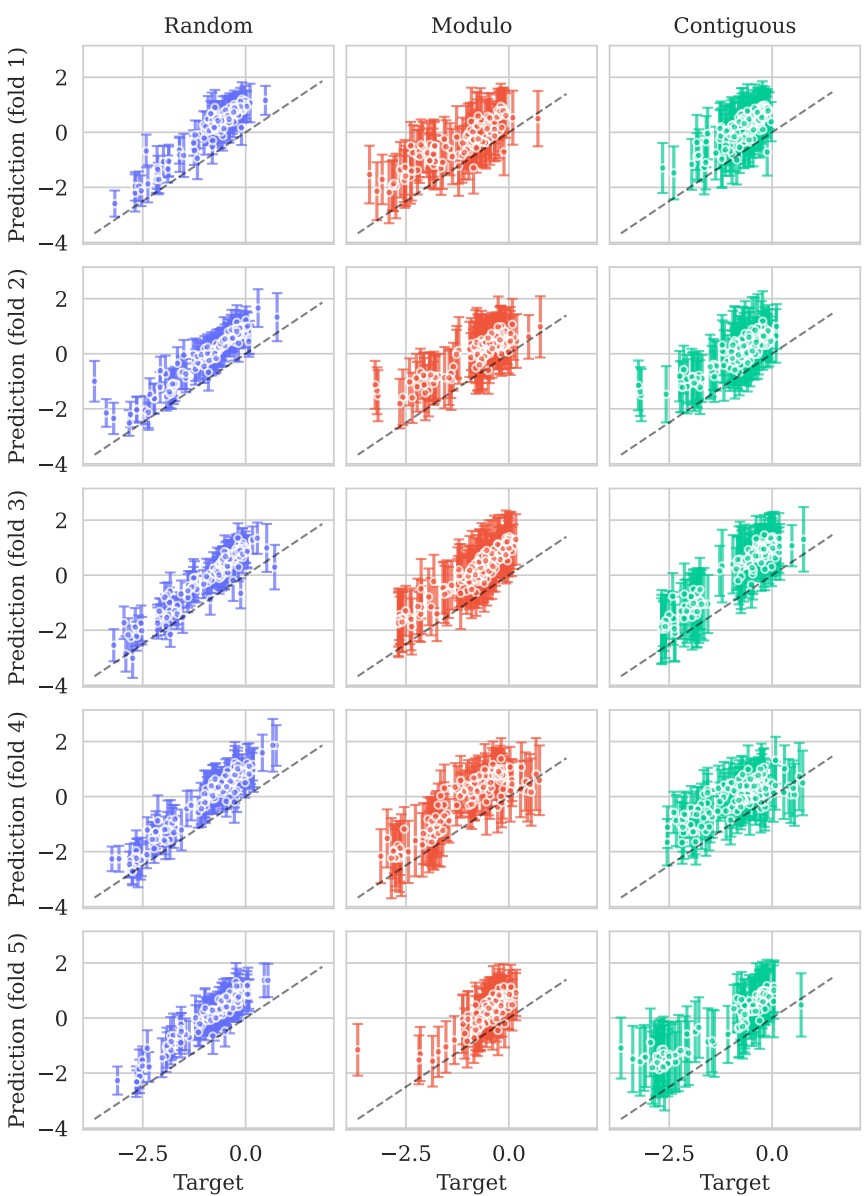

Figure K.3: Predicted means ($\pm 2\sigma$) vs. true values. Columns correspond to CV-schemes. Rows correspond to test folds. Perfect prediction corresponds to dashed diagonal line ($x = y$).

## K.4  OPSD_HUMAN_Wan_2019

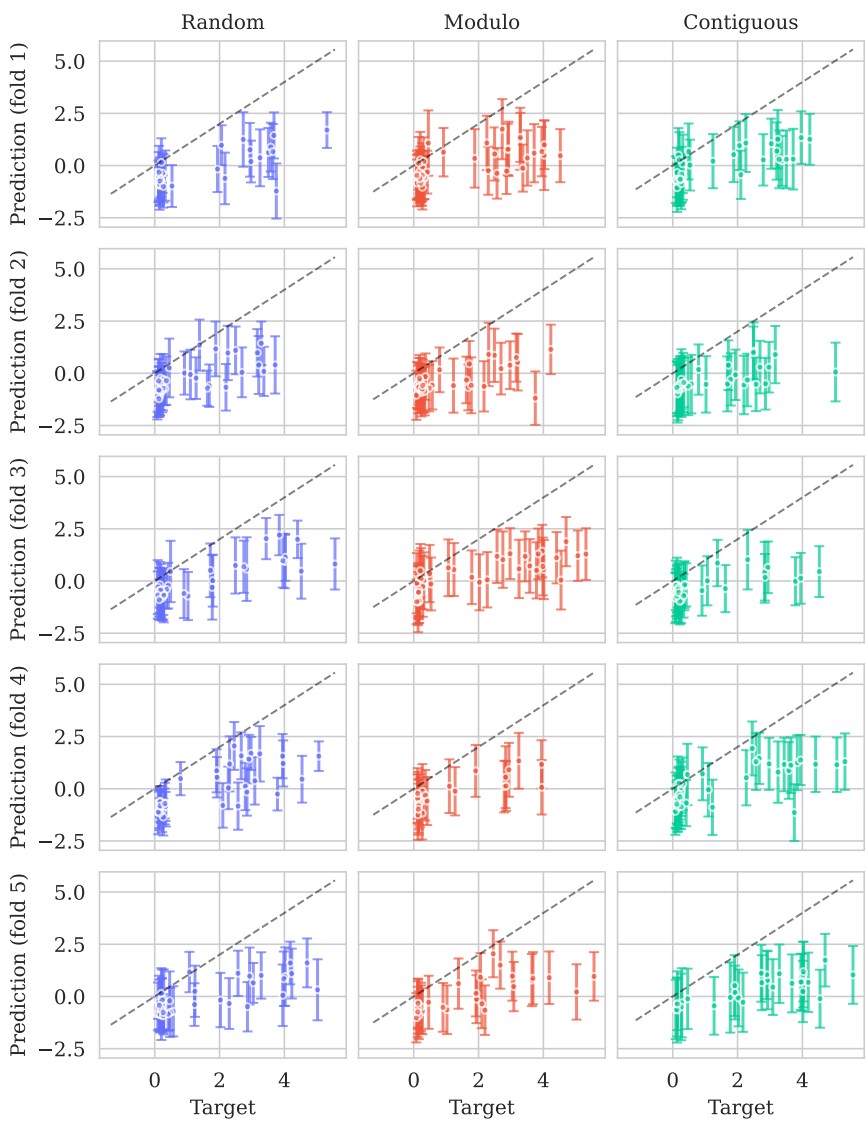

Figure K.4: Predicted means ($\pm 2\sigma$) vs. true values. Columns correspond to CV-schemes. Rows correspond to test folds. Perfect prediction corresponds to dashed diagonal line ($x = y$).

# L Calibration curves for ProteinNPT

## L.1 ProteinNPT details

We use ProteinNPT using the provided software in the paper with the default settings. We generate the MSA Transformer embeddings manually using the provided software from ProteinNPT. During evaluation, we predict using Monte Carlo dropout (with 25 samples, as described in the ProteinNPT appendix). An uncertainty estimate per test sequence is obtained by taking the standard deviation over the 25 samples as prescribed.

## L.2 Calibration curves

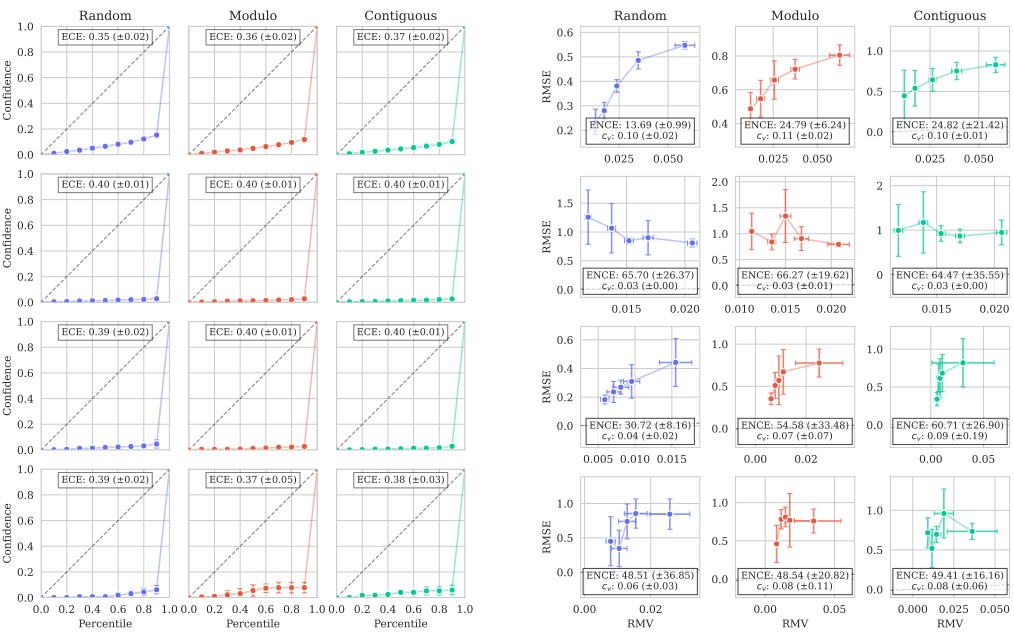

(a) Confidence interval-based calibration curves

(b) Error-based calibration curves

Figure L.1: Calibration curves for ProteinNPT on the four dataset from Table 3. Standard deviation over CV folds is shown as vertical bars. Perfect calibration corresponds to diagonal lines ($y = x$) and is shown as dashed lines in each plot. The predictive uncertainties for ProteinNPT are very small, resulting in poor out-of-the-box calibration as seen on the $x$-axis in (b) and in Figures L.2 to L.5. However, as indicated by the trend in both (a) and (b), the errors correlate with the magnitude of the uncertainties. This suggests that a recalibration might be sufficient to achieve good calibration.

## L.3 Predicted vs. true values for calibration datasets

### L.3.1 BLAT_ECOLX_Stiffler_2015

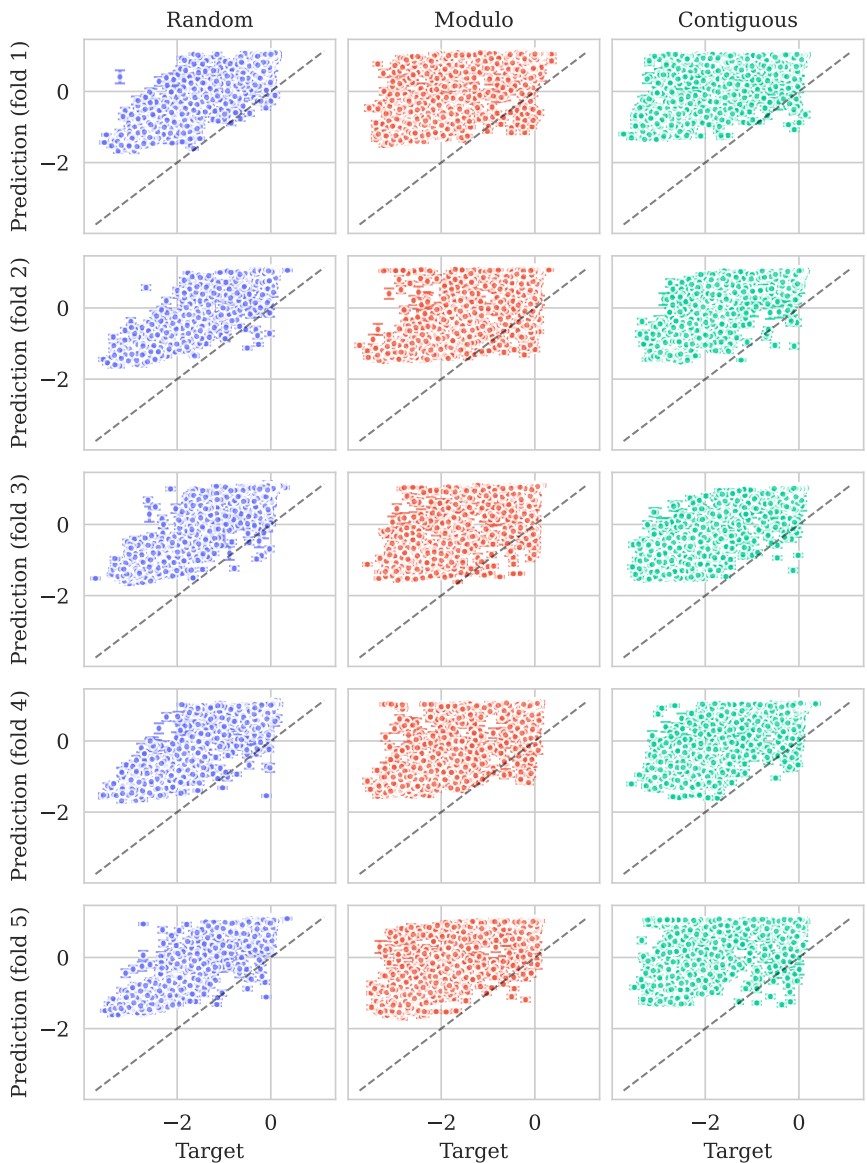

Figure L.2: Predicted means by ProteinNPT ($\pm 2\sigma$) vs. true values. Columns correspond to CV-schemes. Rows correspond to test folds. Perfect prediction corresponds to dashed diagonal line ($x = y$). While the predictions are good, the model is very overconfident.

### L.3.2 PA_I34A1_Wu_2015

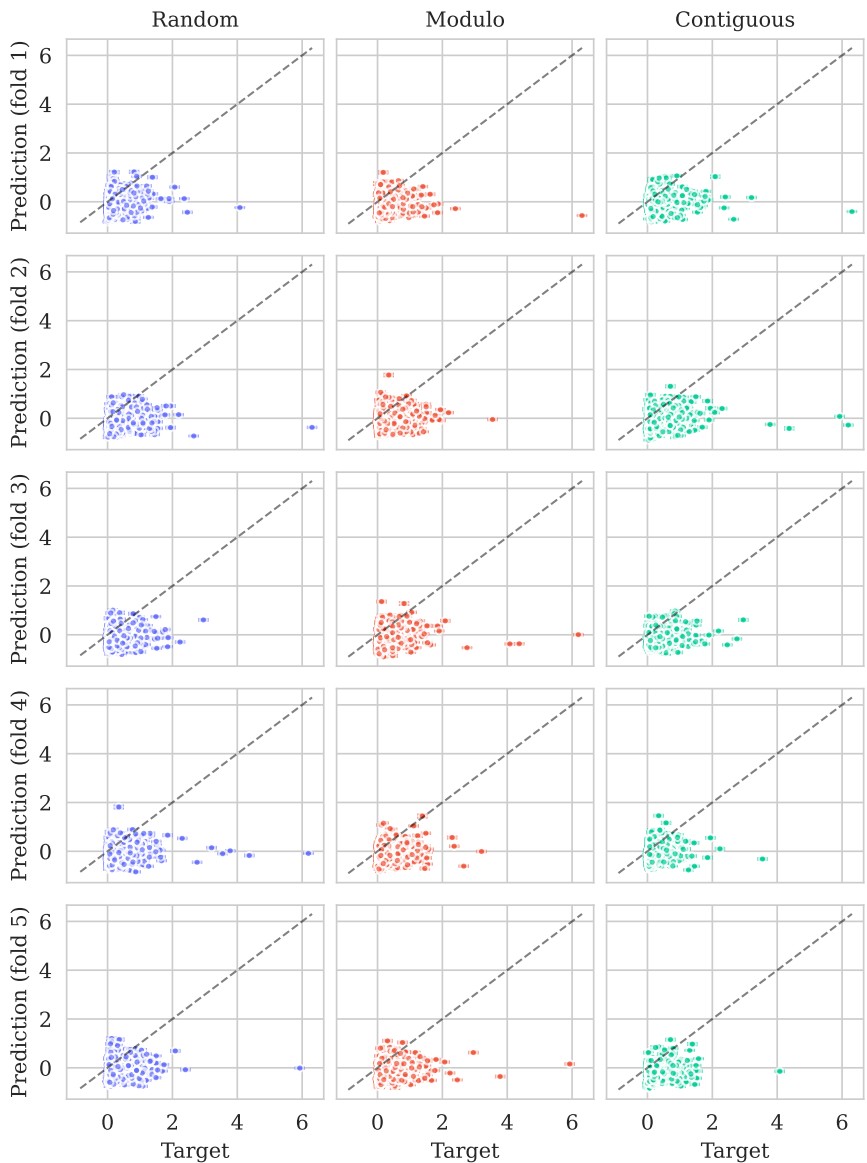

Figure L.3: Predicted means by ProteinNPT ($\pm 2\sigma$) vs. true values. Columns correspond to CV-schemes. Rows correspond to test folds. Perfect prediction corresponds to dashed diagonal line ($x = y$). Despite the relatively poor predictions, the model remains overconfident.

### L.3.3 TCRG1_MOUSE_Tsuboyama_2023

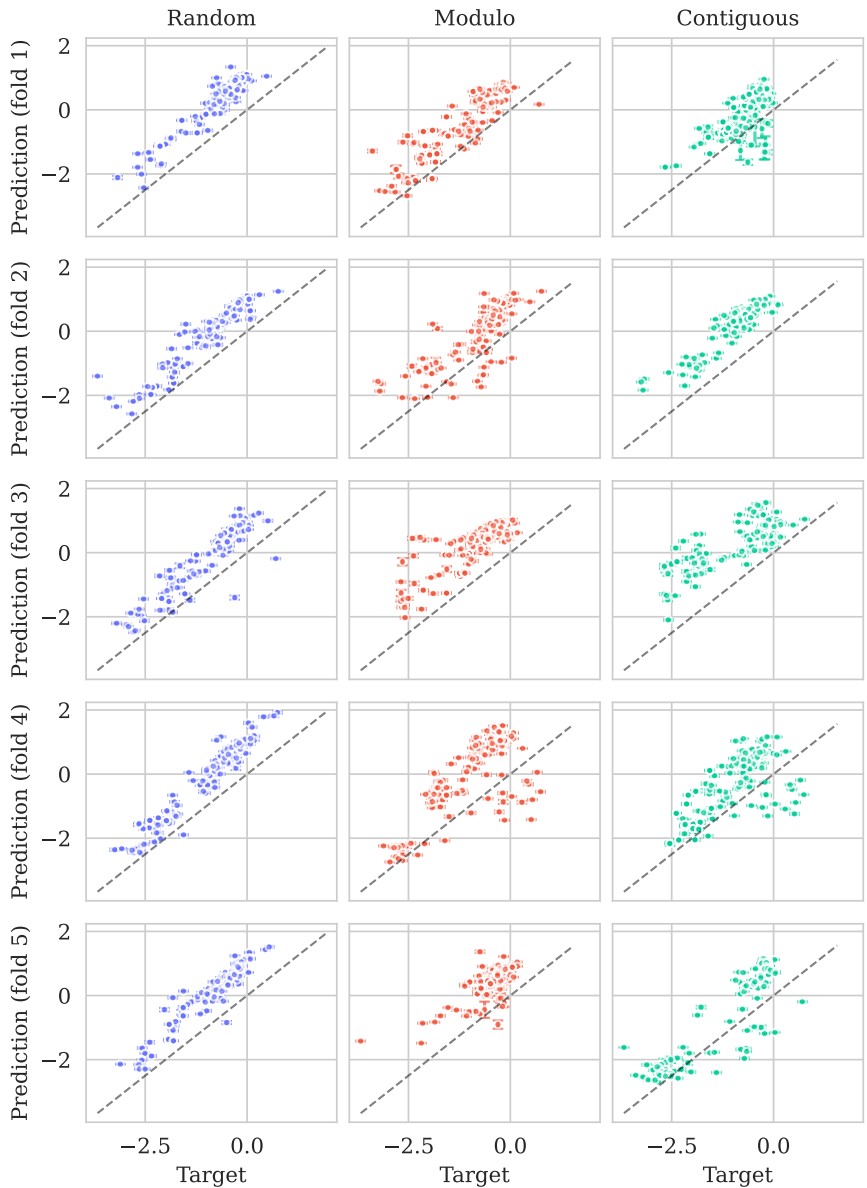

Figure L.4: Predicted means by ProteinNPT ($\pm 2\sigma$) vs. true values. Columns correspond to CV-schemes. Rows correspond to test folds. Perfect prediction corresponds to dashed diagonal line ($x = y$). While the predictions are good, the model is very overconfident. Despite the relatively poor predictions, the model remains overconfident.

**L.3.4   OPSD_HUMAN_Wan_2019**

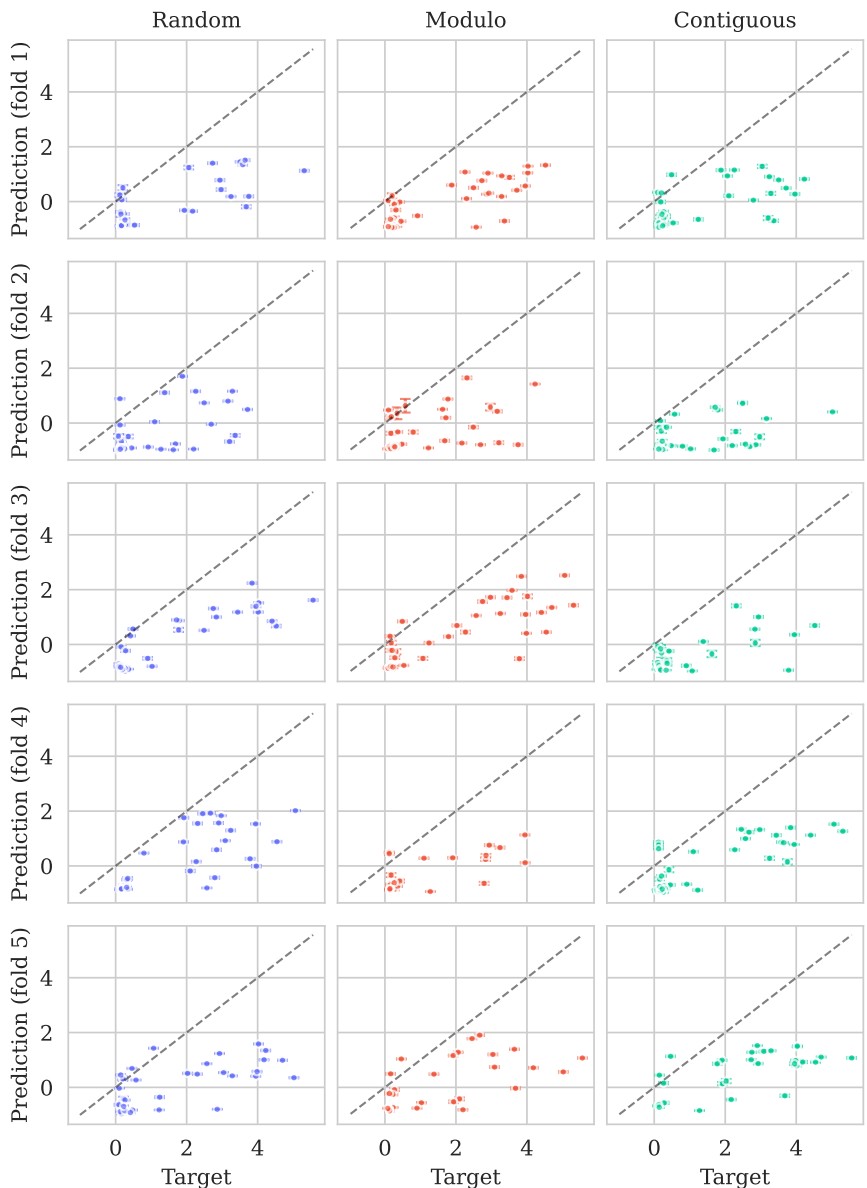

Figure L.5: Predicted means by ProteinNPT ($\pm 2\sigma$) vs. true values. Columns correspond to CV-schemes. Rows correspond to test folds. Perfect prediction corresponds to dashed diagonal line ($x = y$).

# M Alternative zero-shot methods

Kermut uses a linear transformation of a variant's zero-shot score as its mean function. In the main results ESM-2 was used. We here provide additional results where different zero-shot methods are used. The experiments are carried out as the ablation results in Section 4.1, i.e., on 174/217 datasets. All zero-shot scores are pre-computed and are available via the ProteinGym suite.

Using a zero-shot mean function instead of a constant mean evidently leads to increased performance. The magnitude of the improvement depends on the chosen zero-shot method, where the order roughly corresponds to that of the zero-shot scores in ProteinGym. We do however see that opting for EVE yields the largest performance increase.

Table M.1: Performance using alternate zero-shot methods. The experiments are carried out on 174/217 datasets. $^*$: ESM-2 in this table is equivalent to Kermut from the main results in Table 1.

| Zero-shot predictor | Spearman ($\uparrow$) | | | | MSE ($\downarrow$) | | | |
|---|---|---|---|---|---|---|---|---|
| | Contig. | Mod. | Rand. | Avg. | Contig. | Mod. | Rand. | Avg. |
| EVE | 0.608 | 0.627 | 0.750 | 0.662 | 0.731 | 0.682 | 0.412 | 0.608 |
| ESM-2$^*$ | 0.605 | 0.628 | 0.743 | 0.659 | 0.730 | 0.683 | 0.420 | 0.611 |
| GEMME | 0.605 | 0.622 | 0.744 | 0.657 | 0.728 | 0.682 | 0.416 | 0.609 |
| VESPA | 0.606 | 0.623 | 0.742 | 0.657 | 0.737 | 0.698 | 0.424 | 0.620 |
| TranceptEVE L | 0.600 | 0.619 | 0.744 | 0.654 | 0.741 | 0.693 | 0.420 | 0.618 |
| ESM-IF | 0.583 | 0.606 | 0.738 | 0.642 | 0.757 | 0.708 | 0.424 | 0.630 |
| ProteinMPNN | 0.575 | 0.599 | 0.734 | 0.636 | 0.769 | 0.718 | 0.429 | 0.639 |
| Constant mean | 0.569 | 0.596 | 0.735 | 0.633 | 0.770 | 0.718 | 0.429 | 0.639 |

# N Hyperparameter visualization

## N.1 Hyperparameter distributions

The distributions of Kermut's hyperparameters across a number of datasets from ProteinGym can be seen divided by split scheme in Figure N.1. $\lambda$ is a scale parameter for the structure kernel, while $\pi$ is a balancing parameter which lets the model assign importance to the structure and sequence kernels, respectively. $\gamma_1$, $\gamma_2$, and $\gamma_3$ are scale coefficients in the kernels' exponents. Their *inverses* are shown to facilitate easier comparison with the squared exponential kernel's lengthscale, $l_{SE}$.

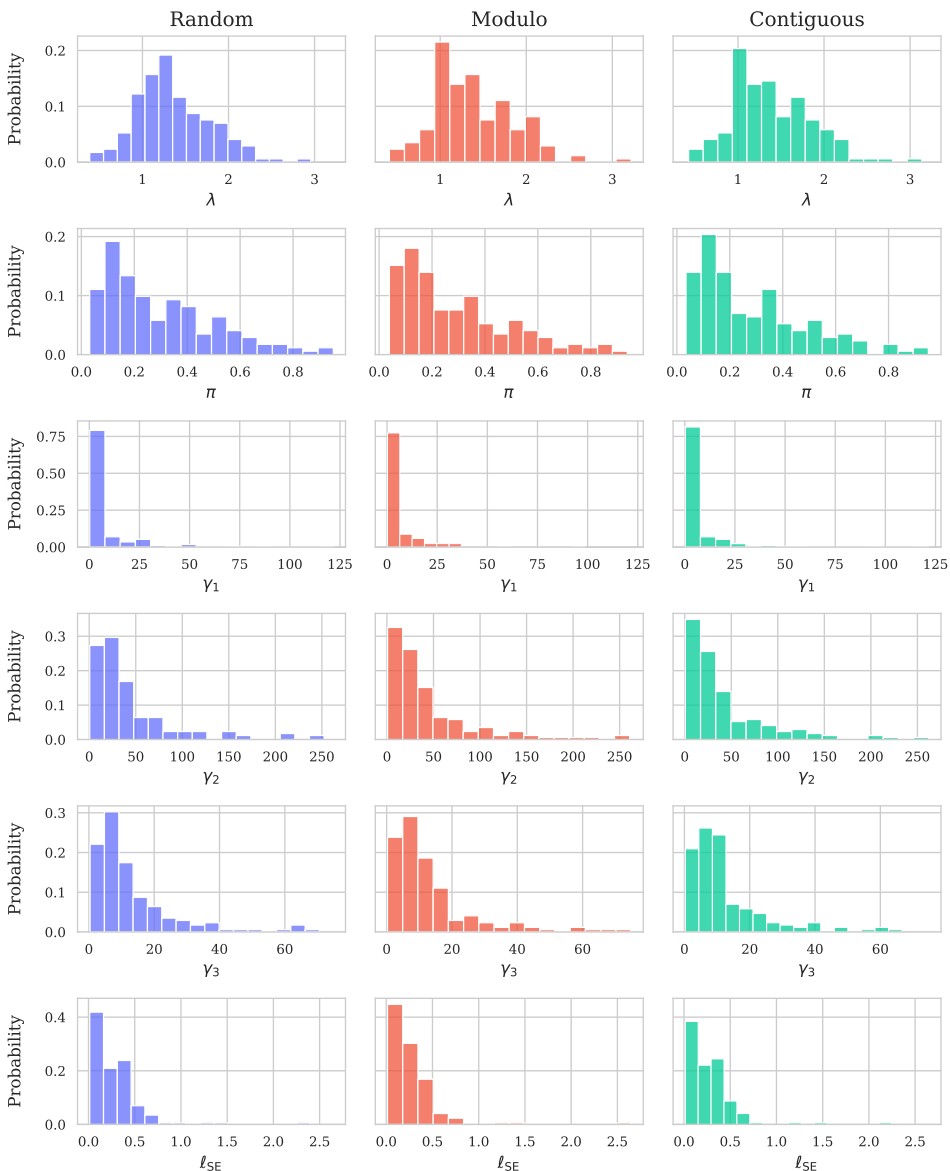

Figure N.1: Distributions of Kermut's hyperparameter across ProteinGym assays and splits. The inverses of $\gamma_1$, $\gamma_2$, and $\gamma_3$ are shown to facilitate easier comparison with the sequence kernel's lengthscale.

## N.2    Hyperparameters vs. dataset size

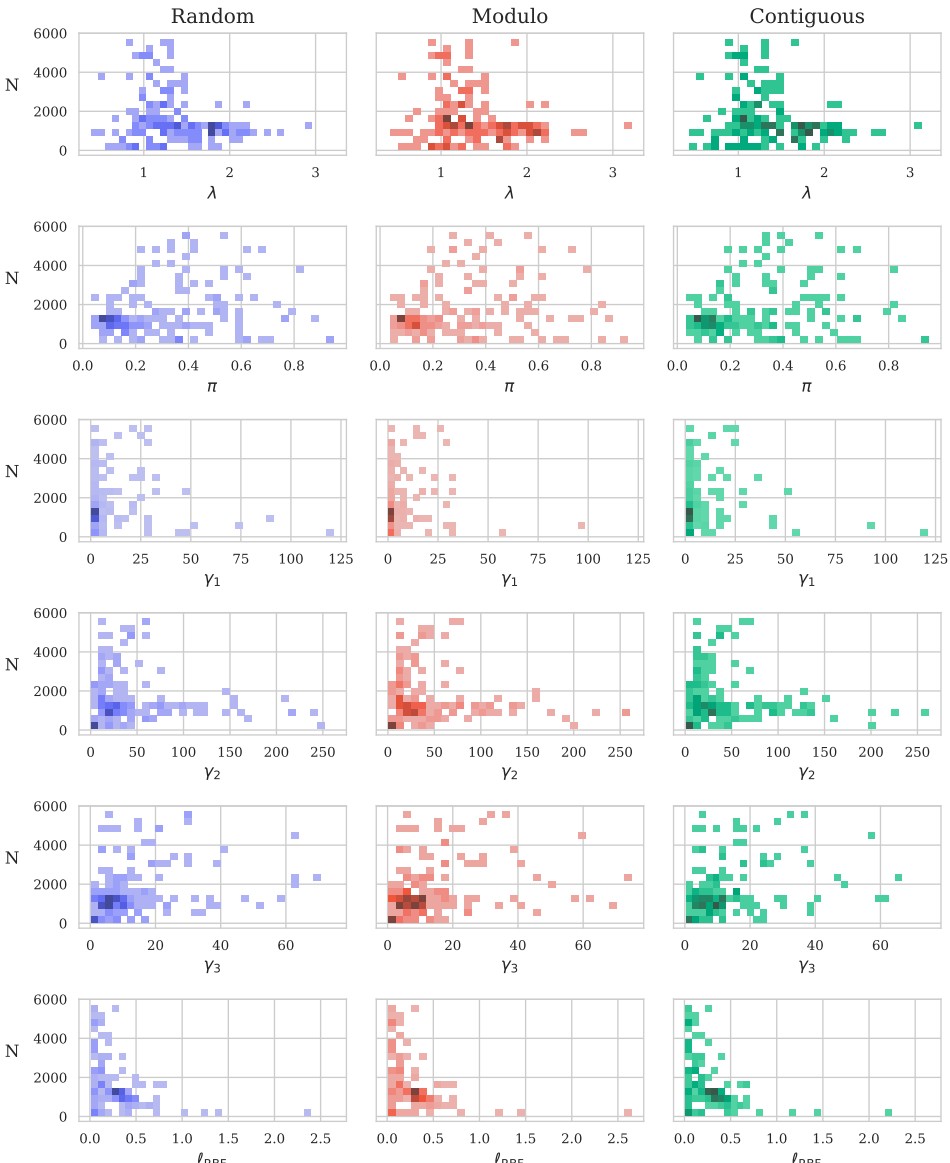

Figure N.2: Distributions of Kermut's hyperparameter across ProteinGym assays and splits visualized against dataset sizes. The inverses of $\gamma_1$, $\gamma_2$, and $\gamma_3$ are shown to facilitate easier comparison with the sequence kernel's lengthscale.

## N.3 Hyperparameters vs. sequence length

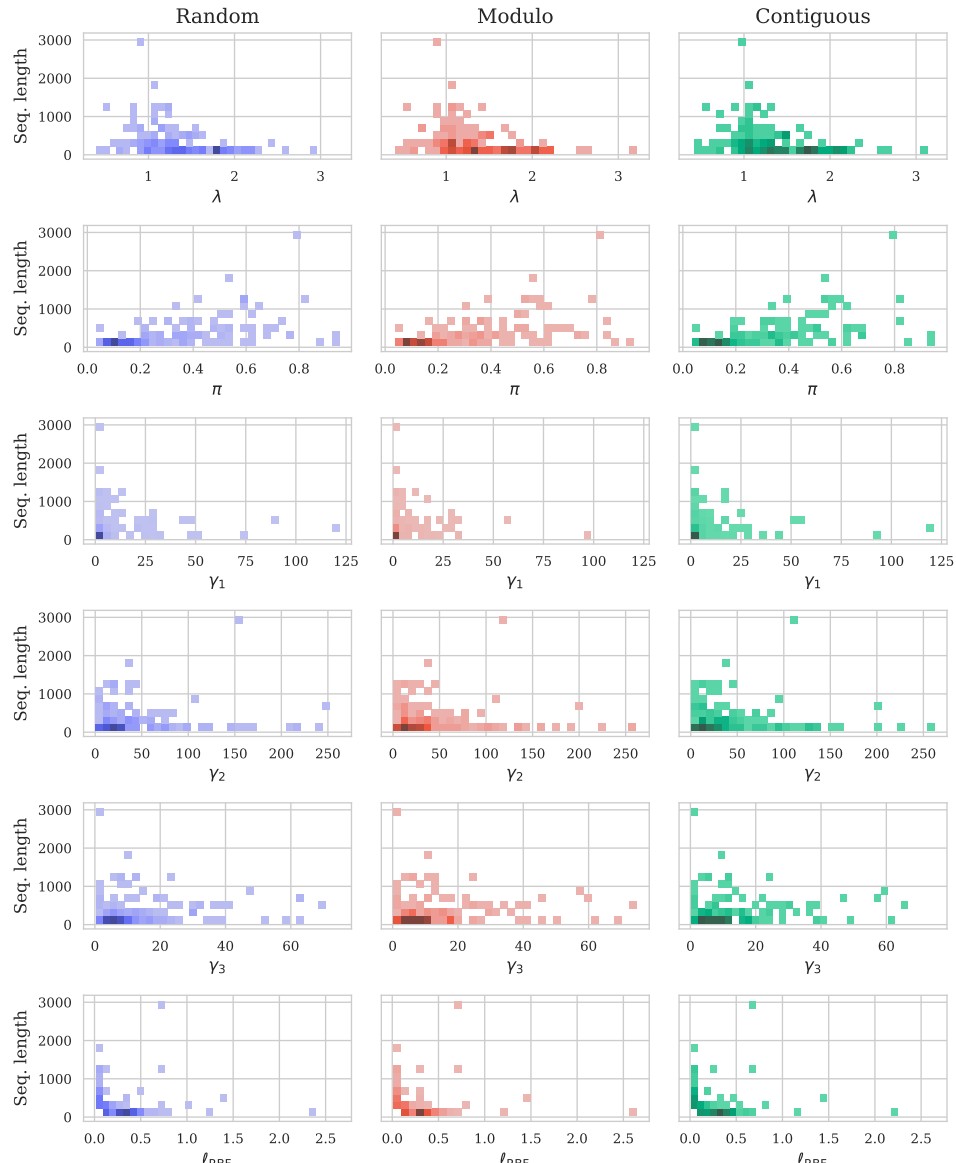

Figure N.3: Distributions of Kermut's hyperparameter across ProteinGym assays and splits visualized against sequence length. The inverses of $\gamma_1$, $\gamma_2$, and $\gamma_3$ are shown to facilitate easier comparison with the sequence kernel's lengthscale.

**N.4   $\pi$ vs. $\lambda$**

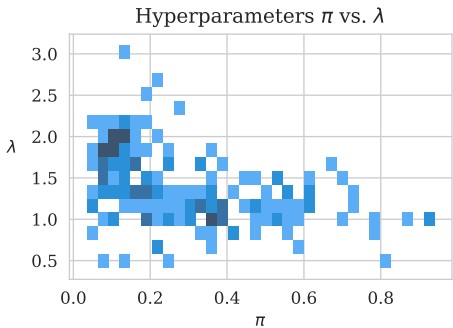

Figure N.4: The structure kernel scale hyperparameter, $\lambda$, is shown against the kernel balancing hyperparameter, $\pi$.

# O  Detailed results per DMS

We show the performance of Kermut and ProteinNPT per DMS assay in Figures O.1 to O.4. The figures show the average performance and the performance per split, respectively.

## O.1  Detailed results per DMS (average)

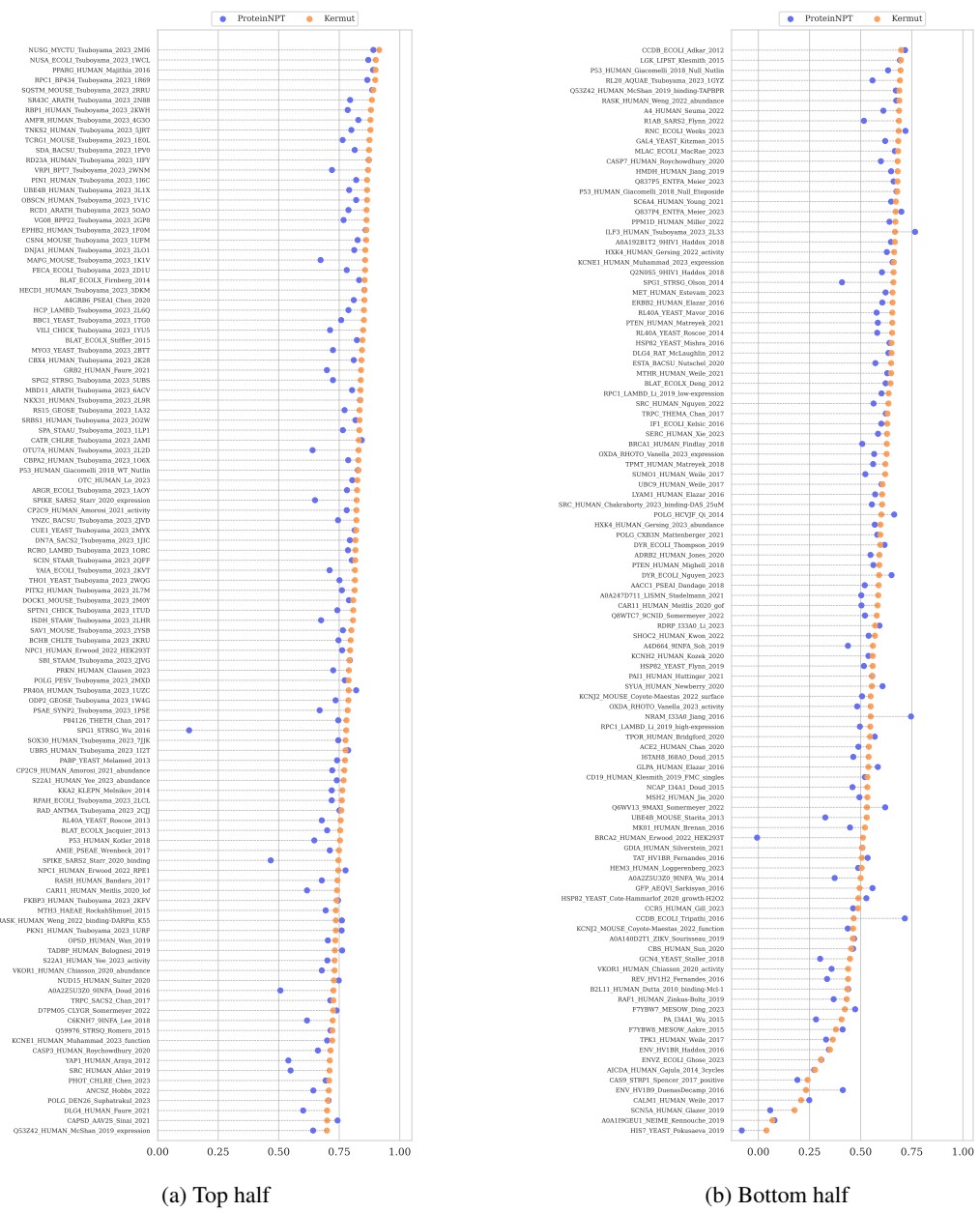

(a) Top half

(b) Bottom half

Figure O.1: Spearman's correlation per DMS assay, averaged over the three split schemes. The figure has been split in two to fit.

## O.2 Detailed results per DMS (random)

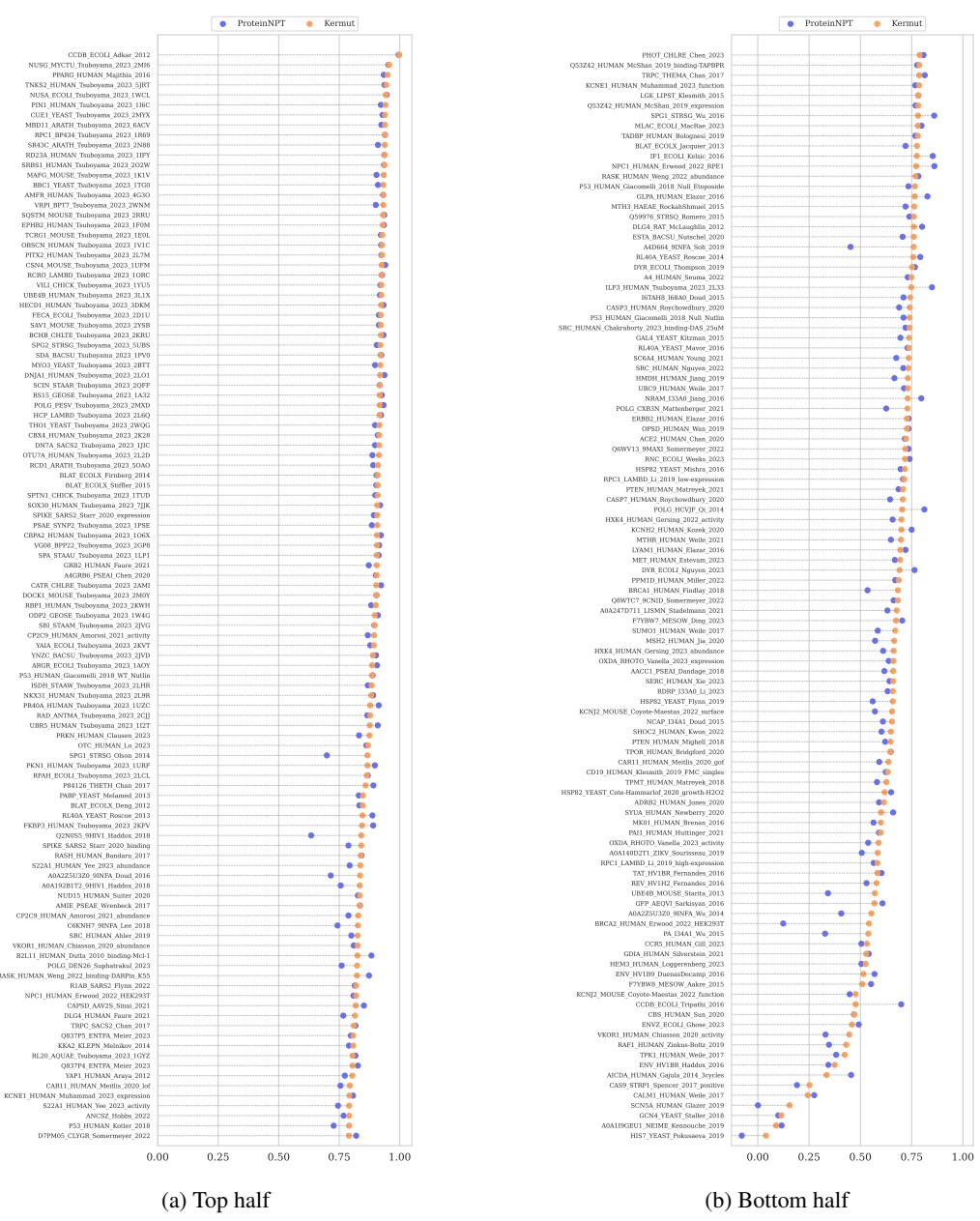

(a) Top half

(b) Bottom half

Figure O.2: Spearman's correlation per DMS assay in the random split scheme. The figure has been split in two to fit. The performance difference between Kermut and ProteinNPT for the random split is smallest (a), i.e., for the assays where the performance is high.

## O.3    Detailed results per DMS (modulo)

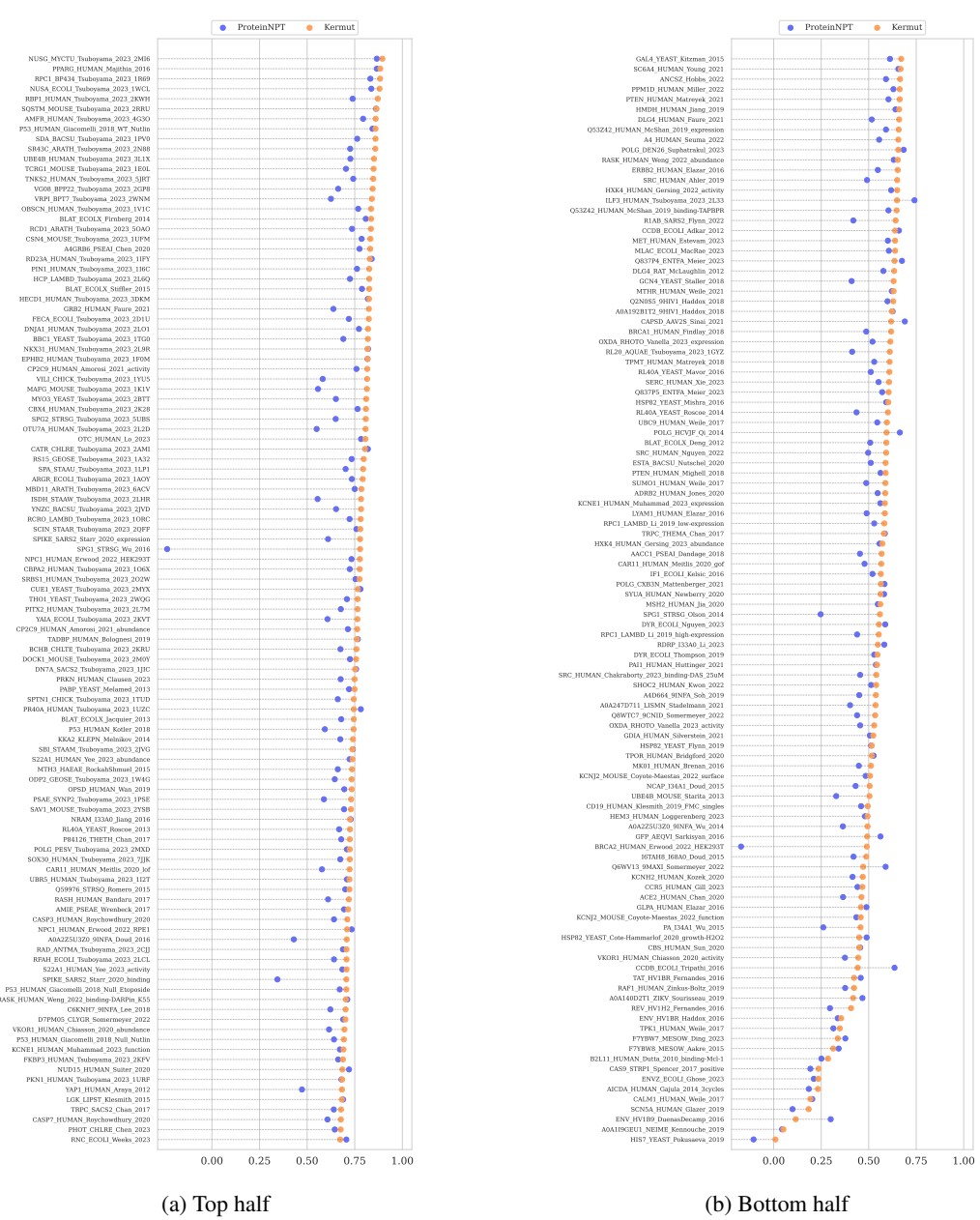

(a) Top half

(b) Bottom half

Figure O.3: Spearman's correlation per DMS assay in the modulo split scheme. The figure has been split in two to fit. In the modulo split we see the clear improvement that the structure kernel offers, where performance increase over ProteinNPT is significantly higher in for many datasets.

## O.4 Detailed results per DMS (contiguous)

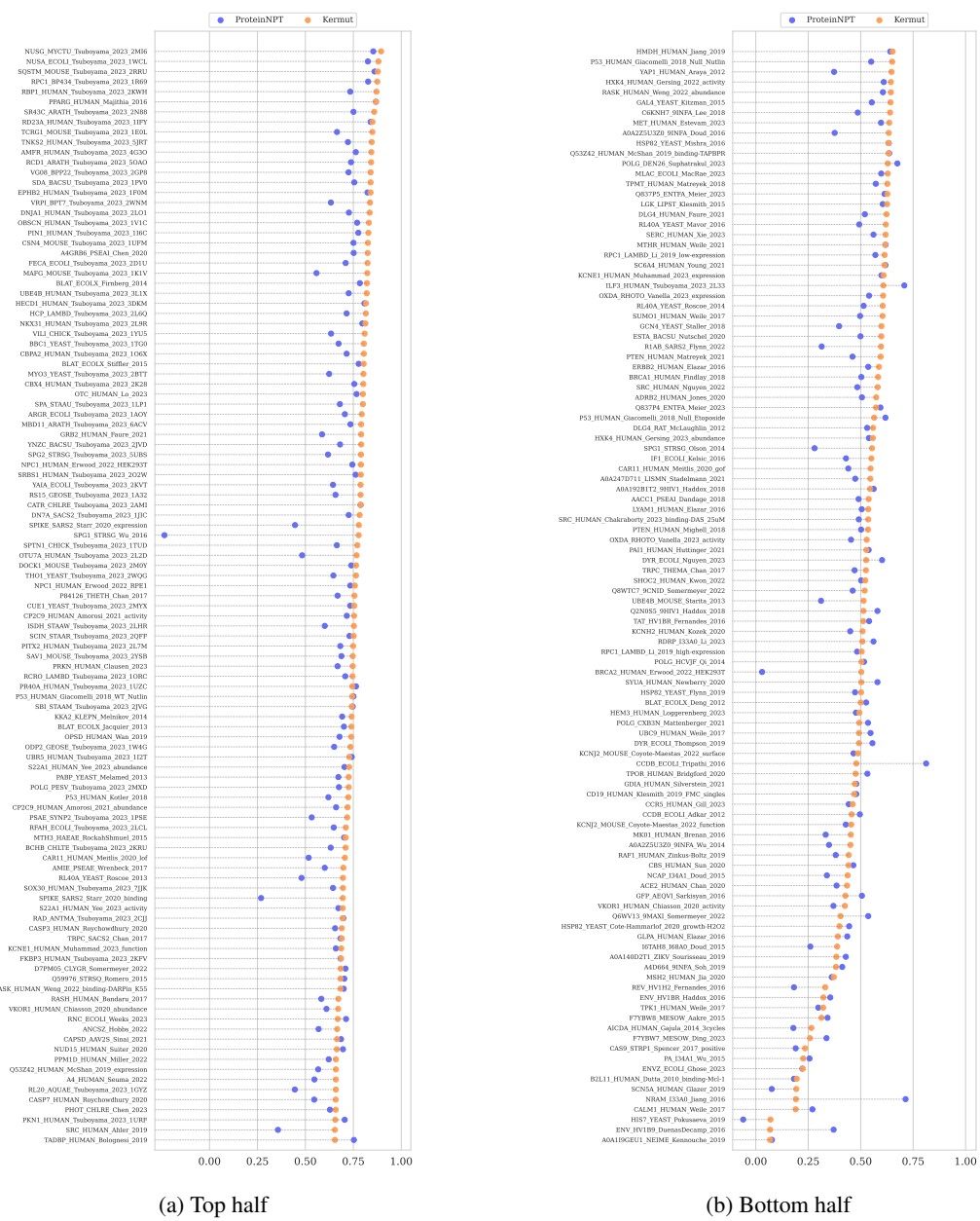

Figure O.4: Spearman's correlation per DMS assay in the contiguous split scheme. The figure has been split in two to fit.

(a) Top half

(b) Bottom half

# P Ethics

We have introduced a general framework to predict variant effects given labeled data. The intent of our work is to use the framework to model and subsequently optimize proteins. We acknowledge that – in principle – any protein property can be modeled (depending on the available data), which means that potentially harmful proteins can be engineered using our method. We encourage the community to use our proposed method for beneficial purposes only, such as the engineering of efficient enzymes or for the characterization of potentially pathogenic variants for the betterment of biological interpretation and clinical treatment.

