# OpenReview forum: "Kermut: Composite kernel regression for protein variant effects"
_NeurIPS.cc/2024/Conference — NeurIPS 2024 spotlight_

### Official Review · Reviewer_Fowf · 2024-07-01

**Soundness:** 3
**Presentation:** 2
**Contribution:** 3
**Rating:** 7
**Confidence:** 4

**Summary:**

The authors introduce a family of Gaussian process based regression models for protein variant effect prediction. The "composite" kernel introduced makes use of structural information (i.e. closeness in 3d space) as well as pre-trained sequence and/or structure models like ESM2 and ProteinMPNN (via embeddings and/or amino acid preference distributions). The model shows promising predictive performance on the ProteinGym benchmark, including w.r.t. uncertainty metrics like expected calibration error (ECE).

**Strengths:**

- The authors contribute to a problem class that is of considerable interest to a fairly large slice of the NeurIPS community, inasmuch as it brings together a compelling application of ML to proteins, touches on transfer learning and representation learning, investigates uncertainty quantification in a difficult problem setting, and leverages classical methods like Gaussian processes
- By using the (ever-larger) ProteinGym benchmark and including quite a few ablations, the authors provide a relatively comprehensive empirical evaluation of their method.
- The performance of the proposed method appears to be pretty good (at least for the regime where you only extrapolate out a few mutations), and could presumably improve as other sequence and/or structure models are plugged-in to the same general construction

**Weaknesses:**

- The description and discussion of the Kermut kernel is not very easy to follow and could be considerably improved.
  - For example, much of the discussion on lines 128-144 seems a bit besides the point, since the authors do not in the end develop a model that is linear in one-hot-features.
  - The authors should do a much better job of foreshadowing/signposting that their construction "for single mutant variants" is but a stepping stone to a multi-mutant construction; otherwise the reader is liable to get confused about what's going on.
  -  Many of the choices that lead to the final kernel construction are either not motivated or only briefly discussed. Why not use, for example, a product kernel $k_{\rm struct} \times k_{\rm seq}$? Some of these alternative choices should be discussed and, ideally, included in ablations.
  - What does this mean? [Line 167] "preventing the comparison of different mutations at the same site"
- The paper is missing a discussion of the computational complexity of computing the kernel w.r.t. the sequence length, the number of mutations from the wildtype, etc.
- The discussion of prior work is inadequate, especially w.r.t. work on sequence kernels and previous applications of GPs to protein modeling. Granted there is only so much space in the main text, but the reader deserves a more detailed discussion. (Perhaps some of the discussion can be relegated to the appendix.) To name just a few examples, there is a lot of work on sequence kernels (see e.g. [A] and references therein) and it is negligent to ignore this body of work. Also the method comparison to mGPfusion [24] is inadequate. The authors did not invent sequence kernels or pioneer their application to protein modeling (e.g. [B, C, D] for more recent work) and should be much more liberal and informative in their discussion of the literature. Granted the relevant literature can be scattered (arxiv, biorxiv, etc.), but the readers deserve (much) better. Discussion of relevant work is more than a required chore: if well done it adds significant value to the reader and the literature as a whole.

References:
- [A] "Biological Sequence Kernels with Guaranteed Flexibility," Alan Nawzad Amin, Eli Nathan Weinstein, Debora Susan Marks, 2023.
- [B] Moss, Henry, et al. "Boss: Bayesian optimization over string spaces." Advances in neural information processing systems 33 (2020): 15476-15486.
- [C] Parkinson, Jonathan, and Wei Wang. "Scalable Gaussian process regression enables accurate prediction of protein and small molecule properties with uncertainty quantitation." arXiv preprint arXiv:2302.03294 (2023).
- [D] Greenman, Kevin P., Ava P. Amini, and Kevin K. Yang. "Benchmarking uncertainty quantification for protein engineering." bioRxiv (2023): 2023-04.

**Questions:**

- Have the authors considered pooling strategies other than mean-pooling?
- What are typical kernel hyperparameters learned across the ProteinGym benchmark? $\gamma_1$, $\pi$, etc. It would be great to see a histogram or similar.
- The throwaway comment [Line 316] "While this limitation can be fixed by online protein structure prediction, the computational cost would increase significantly" seems a bit misleading without further clarification, since kernels are "pairwise" objects and so it's not immediately clear how you would incorporate multiple structures into your kernel. Can you please comment?

**Limitations:**

To my mind the major limitation of this work is that reliance on the ProteinGym benchmark means that there is little signal on how well this class of models would perform in more challenging problem settings where the model is asked to extrapolate out many mutations away from the wild type. There are increasingly many public datasets that make this kind of evaluation possible in principle, and I encourage the authors to apply their method to more challenging scenarios.

For example:
- Chinery, Lewis, et al. "Baselining the Buzz. Trastuzumab-HER2 Affinity, and Beyond!." bioRxiv (2024): 2024-03.

---

> ### Author Rebuttal · Authors · 2024-08-06
>
> We would like to thank the reviewer for their thorough insights and particularly the raised issues regarding the related works section as well as their comments on the methods section.
>
> **Weaknesses:**
> - "The description and discussion..."
>   - "For example, ..."
>   - "The authors should..."
>
> We acknowledge that section 3.3 was not as straightforward to follow as we intended and have modified it accordingly. We have essentially removed lines 128-144, which were meant to describe how the initial kernel came about. We now instead motivate why a structure kernel could be useful and that we intend to introduce a single-mutant kernel which we will later extend to multiple mutations. We then introduce the three sub-kernels sequentially, motivating the inclusion of each. We then mention the lack of epistasis, and use this and the existing literature on GPs on embeddings to motivate the addition of the sequence kernel to the structure kernel.
>
>   - "Many of the choices..."
>
> Reviewer RDFU raised similar points in both weaknesses and questions and due to character constraints, we refer to our responses to this reviewer.
>
>   - "What does this ..."
>
> We agree that the formulation was confusing and have modified it accordingly. Our point is that $k_H$ is incapable of distinguishing between different mutations at the same site since $d_H(x,x')=1$, when $x$ and $x'$ have mutations on the same site.
>
> - "The paper is missing a discussion of the computational complexity..."
>
> This is a good point. We’ve now added information on computational complexity in the appendix w.r.t sequence length, number of mutations, and number of datapoints. Briefly put, assuming precomputed embeddings and AA-distributions, evaluating $k(x,x’)$ has complexity $m_1\times m_2$ (for # mutations in $x$ and $x’$, respectively). The main bottleneck lies in the quadratic scaling of the transformer-based ESM model with sequence length, which can however be substituted with different architectures.
>
> - "The discussion of prior work..."
>
> Thank you for raising this important issue. We agree that our initial related work section was insufficient with respect to the existing literature for both kernel methods on protein sequences and uncertainty quantification and calibration. We have now revised section 2.3 on kernel methods significantly. We have furthermore added a new section on uncertainty quantification and calibration. We have additionally revised the section on local environments to more broadly capture the literature of machine learning modeling on structural environments. For transparency, we will include the revised texts as a comment to this rebuttal (due to the character limit) with included references.
>
> **Questions:**
>
> - “Have the authors considered pooling strategies other than mean-pooling?”
>
> Yes. In Table 2 in "Learning meaningful representations of protein sequences" [Detlefsen et al] they show that alternative pooling strategies can improve the predictive performance on downstream tasks, with significant gains shown by employing a bottleneck approach and to a lesser extent a concatenation as opposed to mean-pooling. While we would expect better performance using a bottleneck approach, this would require the additional computational overhead of training the bottleneck model. This would additionally decrease the flexibility of our model if different embeddings were used as the bottleneck would need to be retrained. A concatenation approach could be employed, but this would lead to significantly different embedding sizes as the protein sequences in ProteinGym range from 37 to thousands of amino acids, which would further complicate the modeling setup.  One can interpret the addition of the structure kernel as a means of recapturing and emphasizing the local sequence differences between variants, which might be less conserved in the embeddings after the averaging operation. We will describe these considerations in the camera-ready version.
>
> - “What are typical kernel hyperparameters learned across the ProteinGym benchmark? , , etc. It would be great to see a histogram or similar.”
>
> We agree! In the global rebuttal, we have uploaded a one-page PDF with visualizations of the hyperparameters. We show histograms for each kernel hyperparameter as well as hyperparameter vs. sequence length and number of variants for 174 ProteinGym datasets. We also show $\lambda$ vs. $\pi$ which shows whether the emphasis is on the structure or sequence kernel (or both). Due to the 1-page constraints, we have only uploaded aggregated results. We will look further into quantitative and qualitative analyses of the hyperparameter distributions for ProteinGym in the coming months.
>
> - “The throwaway comment...”
>
> We agree that the above formulation is unnecessarily vague and even slightly misleading and have now rewritten it to reflect the following: Given a variant with a large number of mutations where we have reasonable suspicion that the structure is different, we would need to predict the structure anew (as well as obtaining AA distributions via ProteinMPNN for the new structure). The difficulty, as you rightly point out, is how we can compare the structure of the multi-mutant variant with, say, a single-mutant variant. The $k_p$ and $k_H$ terms will still be comparable, provided that the local environments are extracted from the different structures. The difficulty lies with the distance kernel, $k_d$. One solution would be to align the two structures, which is $n^3$ in computational complexity, and calculate the distance between the mutants, where the locations are found in their respective structures. Though not ideal, this could make up a possible solution.
>
> **Limitations:**
> - "To my mind..."
>
> We agree that our model needs to be evaluated more thoroughly in multi-mutant settings. We are currently looking into such benchmarks like the suggested paper or the GB1/AAV landscapes from FLIP and will include these in the revised paper.

---

> ### Author Response · Authors · 2024-08-06
> **Revised section 2.2**
>
> *The following is the revised section 2.2 “Kernel methods for protein sequences” (formerly “Kernel methods”), followed by the newly added section 2.3 “Uncertainty quantification and calibration”, and references. We split them into separate comments due to character constraints.*
>
> **2.2 Kernel methods for protein sequences**
>
> Kernel methods have seen much use for protein modeling and specifically protein property prediction. Sequence-based string kernels operating directly on the protein amino acid sequences are one such example, where, e.g., matching k-mers at different ks quantify covariance. This has been used with support vector machines to predict protein homology [24,25]. Another example is sub-sequence string kernels, which in [26] is used in a Gaussian process for a Bayesian optimization procedure. In [27], string kernels were combined with predicted physico-chemical properties to further improve accuracy in the prediction of MHC-binding peptides and in protein fold classification. In [28], a kernel leveraging the tertiary structure for a protein family represented as a residue-residue contact
> map, was used to predict various protein properties such as enzymatic activity and binding affinity. In [29], Gaussian process regression (GPR) was used to successfully identify promising enzyme sequences which were subsequently synthesized showing increased activity. In [30], the authors provide a comprehensive study of kernels on biological sequences which includes a thorough review of the literature as well as both theoretical, simulated, and in-silico results.
>
> Most similar to our work is mGPfusion [31], in which a weighted decomposition kernel was defined which operated on the local tertiary protein structure in conjunction with a number of substitution matrices. Simulated stability data for all possible single mutations were obtained via Rosetta [32], which was then fused with experimental data for accurate ∆∆G predictions of single- and multi-mutant variants via Gaussian process regression, thus incorporating both sequence, structure, and a biophysical simulator. In contrast to our approach, the mGPfusion-model does not leverage pretrained models, but instead relies on substitution matrices for its evolutionary signal. A more recent example of kernel-based methods yielding highly competitive results is xGPR [33], in which GPs with custom kernels show high performance when trained on protein language model embeddings, similarly to the sequence kernel in our work (see Section 3). They use a set of novel random feature-approximated kernels with linear-scaling, where we use the squared exponential kernel. Moreover xGPR does not model structural environments, unlike our structure kernel. Their methods were shown to provide both high accuracy and well-calibrated uncertainty estimation on the FLIP and TAPE benchmarks.

---

> ### Author Response · Authors · 2024-08-06
> **New section 2.3**
>
> **2.3 Uncertainty quantification and calibration**
>
> Uncertainty quantification (UQ) for protein property prediction continues to be a promising area of research with immediate practical consequences. In [34], residual networks were used to model both epistemic and aleatoric uncertainty for peptide selection. In [35], GPR on MLP-residuals from biLSTM embeddings was used to successfully guide in-silico experimental design of kinase binders and protein fluorescence, amongst others. The authors of [36] augmented a Bayesian neural network by placing biophysical priors over the mean function by directly using Rosetta energy scores, whereby the model would revert to the biophysical prior when the epistemic uncertainty was large. This was used to predict fluorescence, binding and solubility for drug-like molecules. In [37], state-of-the-art performance on protein-protein interactions was achieved by using a spectral-normalized neural Gaussian process [38] with an uncertainty-aware transformer-based architecture working on ESM-2 embeddings.
>
> In [39], a framework for evaluating the epistemic uncertainty of deep learning models using confidence interval-based metrics was introduced, while [40] conducted a thorough analysis of uncertainty quantification methods for molecular property prediction. Here, they highlighted the importance of supplementing confidence-based calibration with error-based calibration as introduced in [41], whereby the predicted uncertainties are connected directly to the expected error for a more nuanced calibration analysis. We evaluate our model using confidence-based calibration as well as error-based calibration following the guidelines in [40]. In [42], the authors conducted a systematic comparison of UQ methods on molecular property regression tasks, while [43] investigated calibratedness of regression models for material property prediction. In [44], the above approaches were expanded to protein property prediction tasks where the FLIP [5] benchmark was examined, while [45] benchmarked a number of UQ methods for molecular representation models. In [46], the authors developed an active learning approach for partial charge prediction of metal-organic frameworks via Monte Carlo dropout [47] while achieving decent calibration. In [48], a systematic analysis of protein regression models was conducted where well-calibrated uncertainties were observed for a range of input representations.

---

> ### Author Response · Authors · 2024-08-06
> **References used in section 2.2**
>
> [5] Christian Dallago, Jody Mou, Kadina E. Johnston, Bruce J. Wittmann, Nicholas Bhattacharya, Samuel Goldman, Ali Madani, and Kevin K. Yang. FLIP: Benchmark tasks in fitness landscape inference for proteins, January 2022
>
> [24] Christina Leslie, Eleazar Eskin, and William Stafford Noble. The spectrum kernel: A string kernel for svm protein classification. In Biocomputing 2002, pages 564–575. WORLD SCIENTIFIC, December 2001.
>
> [25] Christina S. Leslie, Eleazar Eskin, Adiel Cohen, Jason Weston, and William Stafford Noble.Mismatch string kernels for discriminative protein classification. Bioinformatics, 20(4):467–476, March 2004.
>
> [26] Henry Moss, David Leslie, Daniel Beck, Javier Gonzalez, and Paul Rayson. Boss: Bayesian optimization over string spaces. Advances in neural information processing systems, 33:15476–15486, 2020.
>
> [27] Nora C Toussaint, Christian Widmer, Oliver Kohlbacher, and Gunnar Rätsch. Exploiting
> physico-chemical properties in string kernels. BMC bioinformatics, 11:1–9, 2010.
>
> [28] Philip A. Romero, Andreas Krause, and Frances H. Arnold. Navigating the protein fitness landscape with Gaussian processes. Proceedings of the National Academy of Sciences, 110(3):E193–E201, January 2013.
>
> [29] Jonathan C Greenhalgh, Sarah A Fahlberg, Brian F Pfleger, and Philip A Romero. Machine learning-guided acyl-acp reductase engineering for improved in vivo fatty alcohol production. Nature communications, 12(1):5825, 2021.
>
> [30] Alan Nawzad Amin, Eli Nathan Weinstein, and Debora Susan Marks. Biological sequence kernels with guaranteed flexibility. arXiv preprint arXiv:2304.03775, 2023.
>
> [31] Emmi Jokinen, Markus Heinonen, and Harri Lähdesmäki. mGPfusion: Predicting protein
> stability changes with Gaussian process kernel learning and data fusion. Bioinformatics,
> 34(13):i274–i283, July 2018.
>
> [32] Andrew Leaver-Fay, Michael Tyka, Steven M Lewis, Oliver F Lange, James Thompson, Ron Jacak, Kristian W Kaufman, P Douglas Renfrew, Colin A Smith, Will Sheffler, et al. Rosetta3: an object-oriented software suite for the simulation and design of macromolecules. In Methods in enzymology, volume 487, pages 545–574. Elsevier, 2011.
>
> [33] Jonathan Parkinson and Wei Wang. Linear-scaling kernels for protein sequences and small molecules outperform deep learning while providing uncertainty quantitation and improved interpretability. Journal of Chemical Information and Modeling, 63(15):4589–4601, 2023.
>
> [34] Haoyang Zeng and David K Gifford. Quantification of uncertainty in peptide-mhc binding
> prediction improves high-affinity peptide selection for therapeutic design. Cell systems, 9(2):159–166, 2019.
>
> [35] Brian Hie, Bryan D. Bryson, and Bonnie Berger. Leveraging Uncertainty in Machine Learning Accelerates Biological Discovery and Design. Cell Systems, 11(5):461–477.e9, November 2020.
>
> [36] Hunter Nisonoff, Yixin Wang, and Jennifer Listgarten. Coherent blending of biophysics-based knowledge with bayesian neural networks for robust protein property prediction. ACS Synthetic Biology, 12(11):3242–3251, 2023.
>
> [37] Young Su Ko, Jonathan Parkinson, Cong Liu, and Wei Wang. Tuna: An uncertainty aware transformer model for sequence-based protein-protein interaction prediction. bioRxiv, pages 2024–02, 2024.
>
> [38] Jeremiah Liu, Zi Lin, Shreyas Padhy, Dustin Tran, Tania Bedrax Weiss, and Balaji Lakshminarayanan. Simple and principled uncertainty estimation with deterministic deep learning via distance awareness. Advances in neural information processing systems, 33:7498–7512, 2020.

---

> ### Author Response · Authors · 2024-08-07
> **References used in section 2.3**
>
> [39] Fredrik K. Gustafsson, Martin Danelljan, and Thomas B. Schön. Evaluating Scalable Bayesian Deep Learning Methods for Robust Computer Vision, April 2020.
>
> [40] Gabriele Scalia, Colin A. Grambow, Barbara Pernici, Yi-Pei Li, and William H. Green. Evaluating Scalable Uncertainty Estimation Methods for Deep Learning-Based Molecular Property Prediction. ACS, 2020.
>
> [41] Dan Levi, Liran Gispan, Niv Giladi, and Ethan Fetaya. Evaluating and Calibrating Uncertainty Prediction in Regression Tasks, February 2020.
>
> [42] Lior Hirschfeld, Kyle Swanson, Kevin Yang, Regina Barzilay, and Connor W Coley. Uncertainty quantification using neural networks for molecular property prediction. Journal of Chemical Information and Modeling, 60(8):3770–3780, 2020.
>
> [43] Kevin Tran, Willie Neiswanger, Junwoong Yoon, Qingyang Zhang, Eric Xing, and Zachary W Ulissi. Methods for comparing uncertainty quantifications for material property predictions. Machine Learning: Science and Technology, 1(2):025006, 2020.
>
> [44] Kevin P Greenman, Ava P Amini, and Kevin K Yang. Benchmarking uncertainty quantification for protein engineering. bioRxiv, pages 2023–04, 2023.
>
> [45] Yinghao Li, Lingkai Kong, Yuanqi Du, Yue Yu, Yuchen Zhuang, Wenhao Mu, and Chao Zhang. Muben: Benchmarking the uncertainty of molecular representation models. Transactions on Machine Learning Research, 2023.
>
> [46] Stephan Thaler, Felix Mayr, Siby Thomas, Alessio Gagliardi, and Julija Zavadlav. Active
> learning graph neural networks for partial charge prediction of metal-organic frameworks via
> dropout monte carlo. npj Computational Materials, 10(1):86, 2024.
>
> [47] Yarin Gal and Zoubin Ghahramani. Dropout as a bayesian approximation: Representing
> model uncertainty in deep learning. In Maria Florina Balcan and Kilian Q. Weinberger,
> editors, Proceedings of The 33rd International Conference on Machine Learning, volume 48 of Proceedings of Machine Learning Research, pages 1050–1059, New York, New York, USA, 20–22 Jun 2016. PMLR.
>
> [48] Richard Michael, Jacob Kæstel-Hansen, Peter Mørch Groth, Simon Bartels, Jesper Salomon, Pengfei Tian, Nikos S. Hatzakis, and Wouter Boomsma. A systematic analysis of regression models for protein engineering. PLOS Computational Biology, 20(5):e1012061, May 2024.

---

> > ### Comment · Reviewer_Fowf · 2024-08-08
> >
> > I thank the authors for engaging in such depth with reviewer feedback. Accordingly I have raised my score to a 7. I look forward to reading the camera ready version of the paper.

---

### Official Review · Reviewer_RDFU · 2024-07-11

**Soundness:** 4
**Presentation:** 2
**Contribution:** 2
**Rating:** 7
**Confidence:** 5

**Summary:**

The authors suggest a method to predict the effect of mutations given sparse data. Their method is based on identifying the similarity of different sites on a protein by embeddings from large language models and structure. They show that their method performs state of the art mutation effect prediction. They also show that their method is only slightly overconfident.

**Strengths:**

The method performs substantially better than state of the art prediction.

The method combines a variety of methods to build a prior on the effect of mutations.

The authors include a codebase that makes this method easy to use for practitioners.

**Weaknesses:**

Epistasis is only included through sequence embeddings; in particular, the impact of structure is purely linear.

Despite not suggesting any radically new technique, this is a practical method that cleverly and effectively uses available tools. For this reason however, I think it is reasonable to expect that the authors try hard to build as strong a method as possible. In particular, this paper is missing a more thorough investigation of model choices -- what if the Hellinger kernel is replaced with something else? what if a different kernel is used to compare embeddings? what if equation 2 is replaced with a kernel with a term for every combination the 3 kernels? what if the kernel is meta-learned on a subset ProteinGym and applied to the rest? The paper would be substantially strengthened by applying the methodology of Duvenaud, David. 2014. “Automatic Model Construction with Gaussian Processes.” University of Cambridge.

**Questions:**

Could you more thoroughly investigate building an accurate kernel method on this data? For example, trying nonlinear structure kernels might help? Using something like $\tilde k(X, X')=\exp(-k(X, X'))$ or an IMQ formulation as in Amin, Alan Nawzad, Eli Nathan Weinstein, and Debora Susan Marks. 2023. “Biological Sequence Kernels with Guaranteed Flexibility.” arXiv [Stat.ML]. arXiv. http://arxiv.org/abs/2304.03775. might be a reasonable thing to try. Or more flexible combinations of kernels as described in the weaknesses section.

**Limitations:**

Partially addressed. I would like a longer discussion about epistasis mentioned in the weaknesses section.

---

> ### Author Rebuttal · Authors · 2024-08-06
>
> We would like to thank the reviewer for their thorough insights and ideas on kernel improvements.
>
> **Weaknesses:**
>
> - “Epistasis is only included through sequence embeddings; in particular, the impact of structure is purely linear.”
>
>
> This is true and to some extent by design. With CoVES, Ding et al. showed impressive results for modeling mutation preferences via simple one-hot encodings and a logistic predictor (and sufficient data). We wondered whether a similar site-specific approach could lead to a performant model, i.e. without explicitly modeling epistasis to keep the model simple, which turned out to be the case. The inclusion of epistatic signals via the sequence embeddings further increased performance, as one would expect. It is certainly possible that the construction of a non-linear global kernel which directly models epistasis could lead to increased performance; particularly in multi-mutant settings. We have updated the method section to describe this.
>
>
> - “Despite not suggesting any radically new technique, this is a practical method that cleverly and effectively uses available tools. For this reason however..."
>
>
> You raise many good questions and suggestions! Prior to submitting our manuscript, our model has undergone many iterations and much testing, including some of the suggestions raised above. For the site-comparison kernel ($k_H$), we have previously experimented with using the Jensen-Shannon divergence (as opposed to the Hellinger distance) and even a simple squared exponential kernel. Here, we saw a minor decrease (non-significant, however) when using the JSD and a larger decrease when using an SE kernel. For the mutation-comparison ($k_p$) and distance kernels ($k_d$), we have experimented with both exponential and squared exponential kernels, however without significant differences in predictive performance on test sets. For the sequence kernel, we experimented with using Mátern kernels (5/2 and 3/2)  which, once again, only led to minor differences. We have now included additional ablation results where we use the JSD (in $k_H$) and Mátern 5/2 (in $k_\text{seq}$).
>
> As for kernel composition, we formulated Kermut as a sum of the structure and sequence kernels due to the additive interpretation given by Duvenaud, where adding kernels is roughly equivalent to a logical OR operation. The model can then leverage either structure or sequence similarities, depending on the presence and strength of each, as determined through the hyperparameter fitting. Using a structured approach such as the Automated Model Construction by Duvenaud might however give rise to a better model. We have now run a set of experiments using this approach with the sum and multiplication operations on our four base kernels (the three kernels from $k_\text{struct}$ and $k_\text{seq}$) on a subset of the ProteinGym benchmark. In each round, we observed increased predictive performance on the respective test sets. The resulting model was a multiplication of all components, i.e.,  $k_\text{struct} \times k_\text{seq}$. We now also include ablation results for this model, where it achieves slightly better performance than our proposed kernel - however within a margin of error (0.662 vs. 0.659 in average Spearman correlation on ablation datasets). For the product kernel, we observe slightly worse calibration in terms of ECE/ENCE with 0.060/0.192 vs. 0.051/0.179 for the original formulation.
>
> **Questions:**
> - “Could you more thoroughly investigate building an accurate kernel method on this data?..."
>
>
> We agree that it would improve the paper to include both richer kernels and more exhaustive baseline comparisons. We have initiated work on implementing the IMQ-H and mGP kernels, and anticipate that these results will be ready in time for the camera-ready version.
>
>
> **Limitations:**
> - “Partially addressed. I would like a longer discussion about epistasis mentioned in the weaknesses section.”
>
>
> Good point. We have now described to what extent epistasis is (and isn’t) incorporated in the model (see the response to your first raised weakness) in the method section and reflect further on it in the discussion.

---

> > ### Comment · Reviewer_RDFU · 2024-08-07
> > **Response**
> >
> > I appreciate you've included a longer discussion of epistasis. I appreciate the results in CoVES and I think you're right that including epistasis is unlikely to substantially increase values on the tested benchmarks. However, I think drawing the conclusion from CoVES that epistasis doesn't matter is not sound; rather I think that this is a result of their benchmark being "hackable". Ultimately, I similarly hope you endeavor to do more than fit single and double mutants in ProteinGym: the sorts of kernels you design could be useful in fitting more diverse data (say from an iterative design experiment) or predicting predicting epistasis as its own end -- in these cases nonlinearity could be more useful.
> >
> > It seems you've done a lot of experiments. I would suggest dumping as many of those results into the appendix as is convenient for you, even if they don't result in major differences.
> >
> > Your additions address my main concerns about the potential to improve the method and I'm happy to recommend the paper be accepted.

---

> > > ### Author Response · Authors · 2024-08-14
> > >
> > > Thank you for you quick response - and the raised score!
> > > We agree that epistasis most likely plays a significant role when multiple mutations are introduced. We will make sure to include additional results in the final paper.

---

### Official Review · Reviewer_1cc6 · 2024-07-12

**Soundness:** 3
**Presentation:** 3
**Contribution:** 3
**Rating:** 7
**Confidence:** 3

**Summary:**

This paper proposes a kernel regression model to predict protein mutational effects. The model includes kernel functions crafted for the task. In specific, it includes a kernel that measures sequence similarity based on ESM-2 features, a local structure similarity kernel based on ProteinMPNN probability, and other kernels that impose priors such as spatial correlation between mutation sites.

**Strengths:**

- Kernels are carefully designed. Using ESM-2 and ProteinMPNN features to construct kernel functions is well-motivated. The effect of each kernel is well-justified via ablation studies (Table 2).
- Uncertainty quantification capability of Gaussian process is valuable for making decisions in wet labs, which has been often neglected in previous work on protein variant effect prediction. This work provides such quantification and further insight into it.
- The model is much faster than purely neural network-based methods which require at least one forward pass per mutation. Kermut is efficient because the sequence and structure features are computed only once for a protein sequence.
- Kermut achieves significant better performance than baselines. It is a good demonstration of making use of pretrained neural network features with statistical methods when training data is not that much and interpretability is desirable.

**Weaknesses:**

- Current formulation of Kermut does not provide transferability to different protein sequences, while previous zero-shot prediction methods are capable of predicting variant effects without prior experimental data on the same sequence.

**Questions:**

- Why is the structure kernel function based on amino acid probability rather than the last-layer feature of ProteinMPNN? And why is the sequence kernel based on last-layer features rather than probability? What is the consideration behind these choices?
- Why is it reasonable to assume that mutations on the same site have similar effects (L135)? It seems that the effects depend more on the amino acid types.

**Limitations:**

See weakness section.

---

> ### Author Rebuttal · Authors · 2024-08-06
>
> We would like to thank the reviewer for their efforts.
>
> **Weaknesses:**
>
> - “Current formulation of Kermut does not provide transferability to different protein sequences, while previous zero-shot prediction methods are capable of predicting variant effects without prior experimental data on the same sequence.”
>
>
> This is true - Kermut only works in an assay-specific, supervised manner. Kermut instead leverages transfer learning from other models (including zero-shot predictions) to drive its performance.
>
>
> **Questions:**
>
> - "Why is the structure kernel function based on amino acid probability rather than the last-layer feature of ProteinMPNN? And why is the sequence kernel based on last-layer features rather than probability? What is the consideration behind these choices?"
>
>
> Good questions! Embeddings extracted from pretrained pLMs like ESM-2 have been shown to be useful for downstream tasks in a number of studies and is thus a straightforward choice of input space to a kernel. We only use the ProteinMPNN output from the wild type protein structure. We could also have chosen to use the last layer node embeddings of ProteinMPNN for each position. This would, however, not include the knowledge of which amino acids the individual sites have been mutated to, but only which site. In other words, there would be no way for the kernel to distinguish between mutations at the same site, since that site would have a high-dimensional vector representation: the information for mutations 73A and 73V would both be represented in this vector for site 73. By using the probability distribution, we have a direct correspondence between elements in our “structure embedding” (the 20 dimensional AA distribution) and the amino acid of interest for a particular variant as modeled by the probability kernel $k_p$. While it is possible to use last-layer embeddings from a structure-based model for the site-comparison kernel ($k_H$) we would need to devise a different kernel for mutation comparisons ($k_p$), which - in our current approach - is handled simultaneously.
> The probability distribution similarly reflects amino acid preferences as shown in the literature. If two sites exhibit high probability for two distinct mutations, these mutations, according to our hypothesis, are likely to lead to a positive (or at least non-negative) change in the functional value. The probabilities therefore offer a convenient way of modeling similarity w.r.t. to function values, which is central to the GP. We have now made this point more clear in the method section.
>
> - "Why is it reasonable to assume that mutations on the same site have similar effects (L135)? It seems that the effects depend more on the amino acid types."
>
>
> If for example an amino acid in the active site of an enzyme is mutated, we expect that most mutations would have a similar negative effect on the activity of the protein. Similarly, we might find that several mutations at a specific site on the surface of a protein might stabilize/destabilize the protein. The amino acid type to which we mutate certainly matters, which is why we model entire sites ($k_H$)  and specific mutations ($k_p$). The paragraph was meant to motivate the choice of kernel, but might have been more confusing and we have thus changed it.

---

### Author Rebuttal · Authors · 2024-08-06

We would first and foremost like to thank all reviewers for their constructive, high-quality reviews. Based on these valuable inputs, we have made a series of alterations to our manuscript, which we will list here. These will be described in greater detail in the individual rebuttals:
- We have made a large revision/expansion of our related works section. We have specifically expanded section 2.2 on kernel methods greatly; we have added a dedicated section to uncertainty quantification and calibration for protein property prediction; and we have expanded the section on local environments to give a more transparent view of the literature. The updated section 2.2 and the new section 2.3 can be seen in full as officialts comments to reviewer Fowf. (Fowf)
- We have changed method section 3.3 (“Kermut”) to give a more concise and straightforward introduction to our kernel, which motivates both how we present it, the individual components, and the kernel composition. (RDFU, Fowf, 1cc6)
- We include in the appendix a model selection section, where we apply an automatic model selection approach on a subset of the ProteinGym datasets (see “Automatic Model Construction with Gaussian Processes” by Duvenaud, 2014) . This gives rise to a $k_{\text{struct}} \times k_{\text{seq}}$-kernel which performs slightly better in predictive performance than our $k_{\text{struct}} + k_{\text{seq}}$-kernel with slightly worse calibration (the performance is within the margin of error). (RDFU)
- We have expanded our ablations to include additional kernel formulations: Mátern 5/2 in sequence kernel, JSD in site-kernel, product of sequence/structure kernel instead of sum. (RDFU, Fowf)
- We have added a discussion of computational complexity in the appendix. (Fowf)
- We have added visualizations of hyperparameters to the appendix. See added PDF for a summarized version. (Fowf)
- We now explicitly mention how we handle and do not handle epistasis in the methods and discussion. (RDFU)

We once again thank the reviewers for their insights and suggestions and believe that the resulting manuscript has been strengthened greatly.

---

### Decision · Program_Chairs · 2024-09-25

**Decision:**

Accept (spotlight)

**Comment:**

There was broad agreement among the reviews that this is a well-written, interesting paper with significant and novel contributions. I fully agree with this positive evaluation, and therefore I recommend acceptance of this work.